# A Best-of-both-worlds Algorithm for Bandits with Delayed Feedback with Robustness to Excessive Delays

**Saeed Masoudian** *
Churney ApS, Denmark
saeed@churney.io

**Julian Zimmert**
Google Research
zimmert@google.com

**Yevgeny Seldin**
University of Copenhagen
seldin@di.ku.dk

## Abstract

We propose a new best-of-both-worlds algorithm for bandits with variably delayed feedback. In contrast to prior work, which required prior knowledge of the maximal delay $d_{\max}$ and had a linear dependence of the regret on it, our algorithm can tolerate arbitrary excessive delays up to order $T$ (where $T$ is the time horizon). The algorithm is based on three technical innovations, which may all be of independent interest: (1) We introduce the first implicit exploration scheme that works in best-of-both-worlds setting. (2) We introduce the first control of distribution drift that does not rely on boundedness of delays. The control is based on the implicit exploration scheme and adaptive skipping of observations with excessive delays. (3) We introduce a procedure relating standard regret with drifted regret that does not rely on boundedness of delays. At the conceptual level, we demonstrate that complexity of best-of-both-worlds bandits with delayed feedback is characterized by the amount of information missing at the time of decision making (measured by the number of outstanding observations) rather than the time that the information is missing (measured by the delays).

## 1 Introduction

Delayed feedback is an ubiquitous challenge in real-world applications. Study of multi-armed bandits with delayed feedback has started at least four decades ago in the context of adaptive clinical trials (Simon, 1977, Eick, 1988), the same problem that has earlier motivated introduction of the bandit model itself (Thompson, 1933). We focus on robustness to delay outliers and to the loss generation mechanism. In practice occasional delay outliers are common (e.g., observations that never arrive). Robustness to the loss generation mechanism implies that the algorithm does not need to know whether the losses are stochastic or adversarial, but still provides regret bounds that match the optimal stochastic rates if the losses happen to be stochastic, while guaranteeing the adversarial rates if they are not (so-called best-of-both-worlds regret bounds). Such algorithms are important from a practical viewpoint, because the loss generation mechanism can rarely assumed to be stochastic, but it is still desirable to have tighter regret bounds if it happens to be. From the theoretical perspective both forms of robustness are interesting and challenging, requiring novel analysis tools and yielding better understanding of the problems.

Joulani et al. (2013) have studied multi-armed bandits with delayed feedback under the assumption that the rewards are stochastic and the delays are sampled from a fixed distribution.

---

*The work was done during SM's employment at the University of Copenhagen.

38th Conference on Neural Information Processing Systems (NeurIPS 2024).

Table 1: Comparison to state-of-the-art. The following notation is used: $T$ is the time horizon, $K$ is the number of arms, $i$ indexes the arms, $\Delta_i$ is the suboptimality gap or arm $i$, $\sigma_{\max}$ is the maximal number of outstanding observations, $D = \sum_{t=1}^{T} d_t$ is the total delay, $\mathcal{S} \subseteq [T]$ is a set of skipped rounds, $\bar{\mathcal{S}} = [T] \setminus \mathcal{S}$ is the set of non-skipped rounds, $D_{\bar{\mathcal{S}}} = \sum_{t \in \bar{\mathcal{S}}} d_t$ is the total delay in the *non*-skipped rounds, and $d_{\max}$ is the maximal delay. We have $\min_{\mathcal{S}} \left( |\mathcal{S}| + \sqrt{D_{\bar{\mathcal{S}}}} \right) \leq \sqrt{D}$ and $\sigma_{\max} \leq d_{\max}$, and in some cases $\min_{\mathcal{S}} \left( |\mathcal{S}| + \sqrt{D_{\bar{\mathcal{S}}}} \right) \ll \sqrt{D}$ and $\sigma_{\max} \ll d_{\max}$.

| Paper | Key results |
|---|---|
| Joulani et al. (2013) | Stochastic bound: $\mathcal{O}\left( \sum_{i:\Delta_i > 0} \left( \frac{\log T}{\Delta_i} + \sigma_{\max} \Delta_i \right) \right)$ |
| Zimmert and Seldin (2020) | Adversarial bound

without skipping: $\mathcal{O}\left( \sqrt{KT} + \sqrt{D \log K} \right)$

with skipping: $\mathcal{O}\left( \sqrt{KT} + \min_{\mathcal{S}} \left( |\mathcal{S}| + \sqrt{D_{\bar{\mathcal{S}}} \log K} \right) \right)$
(Masoudian et al. (2022) provide a matching lower bound) |
| Masoudian et al. (2022)

The results assume oracle

knowledge of $d_{\max}$ | Best-of-both-worlds bound, stochastic part
$\mathcal{O}\left( \sum_{i \neq i^*} \left( \frac{\log T}{\Delta_i} + \frac{\sigma_{\max}}{\Delta_i \log K} \right) + d_{\max} K^{1/3} \log K \right)$
Best-of-both-worlds bound, adversarial part
$\mathcal{O}\left( \sqrt{TK} + \sqrt{D \log K} + d_{\max} K^{1/3} \log K \right)$ |
| Our paper | Best-of-both-worlds bound, stochastic part
$\mathcal{O}\left( \sum_{i \neq i^*} \left( \frac{\log T}{\Delta_i} + \frac{\sigma_{\max}}{\Delta_i \log K} \right) + K\sigma_{\max} + S^* \right)$, where
$S^* = \mathcal{O}\left( \min\left( d_{\max} K^{\frac{2}{3}} \log K, \min_{\mathcal{S}} \left\{ |\mathcal{S}| + \sqrt{D_{\bar{\mathcal{S}}} K^{\frac{2}{3}} \log K} \right\} \right) \right)$
Best-of-both-worlds bound, adversarial part
$\mathcal{O}\left( \sqrt{KT} + \min_{\mathcal{S}} \left\{ |\mathcal{S}| + \sqrt{D_{\bar{\mathcal{S}}} \log K} \right\} + S^* + K\sigma_{\max} \right)$ |

They provided a modification of the UCB1 algorithm for stochastic bandits with non-delayed feedback (Auer et al., 2002). They have shown that the regret of the modified algorithm is $O\left( \sum_{i:\Delta_i > 0} \left( \frac{\log T}{\Delta_i} + \sigma_{\max} \Delta_i \right) \right)$, where $i$ indexes the arms, $\Delta_i$ is the suboptimality gap of arm $i$, $T$ is the time horizon (unknown to the algorithm), and $\sigma_{\max}$ is the maximal number of outstanding observations. (An observation is counted as outstanding at round $t$ if it originates from round $t$ or earlier, but due to delay it was not revealed to the algorithm by the end of round $t$. The number of outstanding observations $\sigma_t$ at round $t$ is the number of actions that have already been played, but their outcome was not observed yet. We also call $\sigma_t$ the [running] count of outstanding observations. The maximal number of outstanding observations $\sigma_{\max}$ is the maximal value that $\sigma_t$ takes and is unknown to the algorithm.) The result implies that in the stochastic setting the delays introduce an additive term in the regret bound, proportional to the maximal number of outstanding observation.

In the adversarial setting, multi-armed bandits with delayed feedback were first analyzed under the assumption of uniform delays (Neu et al., 2010, 2014). For this setting Cesa-Bianchi et al. (2019) have shown an $\Omega(\sqrt{KT} + \sqrt{dT \log K})$ lower bound and an almost matching upper bound, where $K$ is the number of arms and $d$ is a fixed delay. The algorithm of Cesa-Bianchi et al. is a modification of the EXP3 algorithm of Auer et al. (2002b). Cesa-Bianchi et al. used a fixed learning rate that is tuned based on the knowledge of $d$. The analysis is based on control of the drift of the distribution over arms played by the algorithm from round $t$ to round $t + d$. Thune et al. (2019) and Bistritz et al. (2019) provided algorithms for variable adversarial delays, but under the assumption that the delays are known "at action time", meaning that the delay $d_t$ is known at time $t$, when the action is taken, rather that at time $t + d_t$, when the observation arrives. The advanced knowledge of delays was used to tune the learning rate and control the drift of played distribution from round $t$, when an action is played, to round $t + d_t$, when the observation arrives. Alternatively, an advance knowledge of the cumulative delay up to the end of the game could be used for the same purpose. Finally, Zimmert and Seldin (2020) derived an algorithm for the adversarial setting that required no advance knowledge of delays and matched the lower bound of Cesa-Bianchi et al. (2019) within constants. The algorithm and analysis of Zimmert and Seldin avoid explicit control of the distribution drift and are parameterized

by running counts of the number of outstanding observations $\sigma_t$, which is an empirical quantity that is observed at time $t$ ("at the time of action").

Masoudian et al. (2022) attempted to extend the algorithm of Zimmert and Seldin (2020) to the best-of-both-worlds setting. The stochastic part of the analysis of Masoudian et al. is based on a direct control of the distribution drift. The control is achieved by damping the learning rate to make sure that the played distribution on arms is not changing too much from round $t$, when an action is played, to round $t + d_t$, when the loss is observed. Highly varying delays cannot be treated with this approach, because fast learning rates limit the range $d_t$ for which the drift is under control, while slow learning rates prevent learning. Therefore, Masoudian et al. had to reintroduce the assumption that that the maximal delay $d_{\max}$ is known, and used it to tune the learning rate. Unfortunately, damping of the learning rate to control the drift over $d_{\max}$ rounds made $d_{\max}$ show up additively in the bound, meaning that potential presence of even a single delay of order $T$ made both the stochastic and the adversarial bounds linear in the time horizon. We emphasise that the linear dependence of the regret on $d_{\max}$ is real and not an artefact of the analysis, because it comes from damped learning rate.

We introduce a different best-of-both-worlds modification of the algorithm of Zimmert and Seldin (2020) that is fully parameterized by the running count of outstanding observations and requires no advance knowledge of delays or the maximal delay $d_{\max}$. Our algorithm is based on a careful augmentation of the algorithm of Zimmert and Seldin with implicit exploration (described below), followed by application of a skipping technique (also described below) as a tool to limit the time span over which we need to control the distribution shift.

Implicit exploration was introduced by Neu (2015) to control the variance of importance-weighted loss estimates in adversarial bandits. But the exploration parameters add up linearly to the regret bound, making it highly challenging to design a scheme for best-of-both-worlds setting. The implicit exploration schedule of Neu leads to $\Omega(\sqrt{T})$ regret bound and, therefore, unsuitable for that. Jin et al. (2022) introduced a different schedule for adversarial Markov decision processes with delayed feedback. However, it is unknown whether their schedule can work in a stochastic analysis. We introduce a novel schedule and show that it works in best-of-both-worlds setting.

Skipping was introduced by Thune et al. (2019) as a way to limit the dependence of an algorithm on a small number of excessively large delays. The idea is that it is "cheaper" to skip a round with an excessively large delay and bound the regret in the corresponding round by 1, than to include it in the core analysis. Thune et al. have assumed prior knowledge of delays, but Zimmert and Seldin (2020) have perfected the technique by basing it on a running count of outstanding observations. In both works skipping was an optional add-on aimed to improve regret bounds in case of highly unbalanced delays. In our work skipping becomes an indispensable part of the algorithm, because, apart from making the algorithm robust to a few excessively large delays, it also limits the time span over which the control of distribution drift is needed.

In Table 1 we compare our results to state of the art. In a nutshell, we replace terms dependent on $d_{\max}$ by terms dependent on $\sigma_{\max}$, and terms dependent on the square root of the total cumulative delay $D = \sum_{t=1}^{T} d_t$, by terms dependent on the number of skipped rounds $|\mathcal{S}|$ and a square root of the cumulative delay $D_{\bar{\mathcal{S}}} = \sum_{t \in \bar{\mathcal{S}}} d_t$ in the non-skipped rounds $\bar{\mathcal{S}}$ (those with the smaller delay). This yields robustness to excessive delays, because neither $\sigma_{\max}$ nor $\min_{\mathcal{S}} \left(|\mathcal{S}| + \sqrt{D_{\bar{\mathcal{S}}}}\right)$ depend on the magnitude of delay outliers. By contrast, both the stochastic and the adversarial regret bounds of Masoudian et al. (2022) become linear in $T$ in presence of a single delay of order $T$.

There are also additional benefits. It has been shown that $\sigma_{\max} \leq d_{\max}$, and in some cases $\sigma_{\max} \ll d_{\max}$ (Joulani et al., 2013, Masoudian et al., 2022). For example, if the first observation has delay $T$, and the remaining observations have zero delay, then $d_{\max} = T$, but $\sigma_{\max} = 1$. We also have that $\min_{\mathcal{S}} \left(|\mathcal{S}| + \sqrt{D_{\bar{\mathcal{S}}}}\right) \leq \sqrt{D}$, because $\mathcal{S} = \varnothing$ is part of the minimization on the left, and in some cases $\min_{\mathcal{S}} \left(|\mathcal{S}| + \sqrt{D_{\bar{\mathcal{S}}}}\right) \ll \sqrt{D}$. For example, if the delays in the first $\sqrt{T}$ rounds are of order $T$, and the delays in the remaining rounds are zero, then $\min_{\mathcal{S}} \left(|\mathcal{S}| + \sqrt{D_{\bar{\mathcal{S}}}}\right) = \mathcal{O}\left(\sqrt{T}\right)$, but $\sqrt{D} = \Omega\left(T^{3/4}\right)$ (Thune et al., 2019). Therefore, bounds that exploit skipping are preferable over bounds that do not, and for some problem instances the improvement is significant. In Appendix F we show that bounds with an additive term $d_{\max}$, including the results of Masoudian et al. (2022), cannot benefit from skipping, in contrast to ours.

The following list highlights our main contributions.

1. We provide the first best-of-both-worlds algorithm for bandits with delayed feedback that is robust to delay outliers. It improves both the stochastic and the adversarial regret bounds relative to the work of Masoudian et al. (2022), which lacks such robustness. For some problem instances the improvement is dramatic, e.g., in presence of a single delay of order $T$ both the stochastic and the adversarial regret bounds of Masoudian et al. are of order $T$, whereas our bounds are unaffected.

2. We provide an efficient technique to control the distribution drift under highly varying delays.

3. We provide the first implicit exploration scheme that works in best-of-both-worlds setting.

4. We provide a procedure relating drifted regret to normal regret in presence of delay outliers.

5. At the conceptual level, we show that best-of-both-worlds regret depends on the amount of information missing at the time of decision making (the number of outstanding observations) rather than the time that the information is missing (the delays). It was shown to be the case for the stochastic and adversarial regimes in isolation (Joulani et al., 2013, Zimmert and Seldin, 2020), but we are the first to show that it is also the case for best-of-both-worlds.

## 2  Problem setting

We study the problem of multi-armed bandit with variable delays. In each round $t = 1, 2, \ldots$, the learner picks an action $I_t$ from a set of $K$ arms and immediately incurs a loss $\ell_{t,I_t}$ from a loss vector $\ell_t \in [0,1]^K$. However, the incurred loss is observed by the learner only after a delay of $d_t$, at the end of round $t + d_t$. The delays are arbitrary and chosen by the environment. We use $\sigma_t$ to denote the number of outstanding observations at time $t$ defined as $\sigma_t = \sum_{s \le t} \mathbb{1}(s + d_s > t)$ and $\sigma_{\max} = \max_{t \in [T]} \sigma_t$ to be the maximal number of outstanding observations. We consider two regimes for generation of losses by the environment: oblivious adversarial and stochastic.

We use pseudo-regret to compare the expected total loss of the learner's strategy to that of the best fixed action in hindsight. Specifically, the pseudo-regret is defined as:

$$\overline{Reg}_T = \mathbb{E}\left[\sum_{t=1}^T \ell_{t,I_t}\right] - \min_{i \in [K]} \mathbb{E}\left[\sum_{t=1}^T \ell_{t,i}\right] = \mathbb{E}\left[\sum_{t=1}^T \left(\ell_{t,I_t} - \ell_{t,i_T^*}\right)\right],$$

where $i_T^* = \min_{i \in [K]} \mathbb{E}\left[\sum_{t=1}^T \ell_{t,i}\right]$ is the best action in hindsight. In the oblivious adversarial setting, the losses are assumed to be deterministic and independent of the actions taken by the algorithm. As a result, the expectation in the definition of $i_T^*$ can be omitted and the pseudo-regret definition coincides with the expected regret. Throughout the paper we assume that $i_T^*$ is unique. This is a common simplifying assumption in best-of-both-worlds analysis (Zimmert and Seldin, 2021). Tools for elimination of this assumption can be found in Ito (2021).

## 3  Algorithm

The algorithm is a best-of-both-worlds modification of the adversarial FTRL algorithm with hybrid regularizer by Zimmert and Seldin (2020). It is provided in Algorithm 1 display. The modification includes biased loss estimators (implicit exploration) and adjusted skipping threshold. The algorithm maintains a set of skipped rounds $\mathcal{S}_t$ (initially empty), a cumulative count of "active" outstanding observations (those that have not been skipped yet), and a vector of cumulative observed loss estimates $\widehat{L}_t^{obs}$ from non-skipped rounds. At round $t$ the algorithm constructs an FTRL distribution $x_t$ over arms using regularizer $F_t$ defined in equation (2) below, and samples an arm according to $x_t$. Then it receives the observations that arrive at round $t$, except those that come from the skipped rounds, and updates the vector $\widehat{L}_t^{obs}$ of cumulative loss estimates. The loss estimates $\widehat{\ell}_t$ are defined below in equation (1). Then it counts the number of "active" outstanding observations $\widehat{\sigma}_t$ (those that belong to non-skipped rounds), updates the cumulative count of outstanding observations $\mathcal{D}_t$, and computes the skipping threshold $d_{\max}^t = \sqrt{\frac{\mathcal{D}_t}{49K^{2/3}\log K}}$. Finally, it adds rounds $s$ for which the observation

---

**Algorithm 1:** Best-of-both-worlds algorithm for bandits with delayed feedback

---

1    **Initialize** $\mathcal{S}_0 = \varnothing$, $\mathcal{D}_0 = 0$, and $\widehat{L}_0^{obs} = \mathbf{0}$, where $\mathbf{0}$ is the zero vector in $\mathbb{R}^K$

2    **for** $t = 1, 2, \ldots$ **do**

3      // *Playing an arm and receiving observations (except from skipped rounds)*

4      Set $x_t = \arg\min_{x \in \Delta^{K-1}} \langle \widehat{L}_{t-1}^{obs}, x \rangle + F_t(x)$          // *$F_t$ is defined in* (2)

5      Sample $I_t \sim x_t$

6      **for** $s : (s + d_s = t) \wedge (s \notin \mathcal{S}_{t-1})$ **do**

7          Observe $(s, \ell_{s, I_s})$

8          $\widehat{L}_t^{obs} = \widehat{L}_{t-1}^{obs} + \widehat{\ell}_s$          // *$\widehat{\ell}_s$ is defined in* (1)

9      // *Counting "active" outstanding observations and updating the skipping threshold*

10      Set $\widehat{\sigma}_t = \sum_{s \in [t-1] \setminus \mathcal{S}_{t-1}} \mathbb{1}(s + d_s > t)$

11      Update $\mathcal{D}_t = \mathcal{D}_{t-1} + \widehat{\sigma}_t$

12      Set $d_{\max}^t = \sqrt{\mathcal{D}_t / \left(49 K^{\frac{2}{3}} \log K\right)}$

13      // *Skipping observations with excessive delays (by Lemma 20 at most one is skipped at a time)*

14      **for** $s \in [t-1] \setminus \mathcal{S}_{t-1}$ **do**

15          **if** $\min\{d_s, t - s\} \geq d_{\max}^t$ **then**

16              $\mathcal{S}_t = \mathcal{S}_{t-1} \cup \{s\}$      // If the waiting time $t - s$ exceeds $d_{\max}^t$, then $s$ is skipped

17          **else**

18              $\mathcal{S}_t = \mathcal{S}_{t-1}$

---

has not arrived yet and the waiting time $(t - s)$ exceeds the skipping threshold $d_{\max}^t$ to the set of skipped rounds $\mathcal{S}_t$. Lemma 20, which is an adaptation of Zimmert and Seldin (2020, Lemma 5) to our skipping rule, shows that at most one round $s$ is skipped at a time (at most one index $s$ satisfies the if-condition for skipping in Line 15 of the algorithm for a given $t$).

We use implicit exploration to control importance-weighted loss estimates. The idea of using implicit exploration is inspired by the works of Neu (2015) and Jin et al. (2022), but its parametrization and purpose are different from prior work. To the best of our knowledge, it is the first time implicit exploration is used for best-of-both-worlds bounds. For any $s, t \in [T]$ with $s \leq t$ we define implicit exploration terms $\lambda_{s,t} = e^{-\frac{\mathcal{D}_t}{\mathcal{D}_t - \mathcal{D}_s}}$. Our biased importance-weighted loss estimators are defined by

$$\widehat{\ell}_{t,i} = \frac{\ell_{t,i} \mathbb{1}(I_t = i)}{\max\left\{x_{t,i}, \lambda_{t, t + \widehat{d}_t}\right\}}, \tag{1}$$

where $\widehat{d}_s = \min\left(d_s, \min\{(t - s) : t - s \geq d_{\max}^t\}\right)$ denotes the time that the algorithm waits for the observation from round $s$. It is the minimum of the delay $d_s$, and the time $(t - s)$ to the first round when the waiting time exceeds the skipping threshold $d_{\max}^t$.

We use a hybrid regularizer based on a combination of the negative Tsallis entropy and the negative entropy, with separate learning rates,

$$F_t(x) = -2\eta_t^{-1}\left(\sum_{i=1}^K \sqrt{x_i}\right) + \gamma_t^{-1}\left(\sum_{i=1}^K x_i \log x_i\right), \tag{2}$$

where the learning rates are $\eta_t^{-1} = \sqrt{t}$ and $\gamma_t^{-1} = \sqrt{\frac{49\mathcal{D}_t}{\log K}}$. The regularizer is the same as the one used by Zimmert and Seldin (2020). By inheriting their regularizer we inherit their adversarial regret bound, which is minimax optimal, with just a minor adjustment due to introduction of implicit exploration and a slight change in the learning rates and skipping threshold. The main contribution of our work is carrying out the stochastic analysis while staying within the algorithmic framework of Zimmert and Seldin and keeping the adversarial regret bound almost unscathed.

The update rule for $x_t$ is

$$x_t = \nabla \bar{F}_t^*(-\widehat{L}_t^{obs}) = \arg\min_{x \in \Delta^{K-1}} \langle \widehat{L}_t^{obs}, x \rangle + F_t(x), \tag{3}$$

where $\widehat{L}_t^{obs} = \sum_{s=1}^{t-1} \widehat{\ell}_s \mathbb{1}(s + d_s < t)\mathbb{1}(s \notin \mathcal{S}_{t-1})$ is the cumulative importance-weighted loss estimate of observations that have arrived by time $t$ and have not been skipped. We use $\mathcal{S}^* = \mathcal{S}_T$ to denote the final set of skipped rounds at time $T$.

# 4 Regret Bounds

The following theorem provides best-of-both-worlds regret bounds for Algorithm 1. A proof is provided in Section 5 and a bound on $S^*$ can be found in Appendix H.

**Theorem 1.** *The pseudo-regret of Algorithm 1 for any sequence of delays and losses satisfies*

$$\overline{Reg}_T = \mathcal{O}\left( \sqrt{KT} + \min_{\mathcal{S} \subseteq [T]} \left\{ |\mathcal{S}| + \sqrt{\mathcal{D}_{\bar{\mathcal{S}}} \log K} \right\} + S^* + K\widehat{\sigma}_{\max} \right),$$

*where $\widehat{\sigma}_{\max} = \max_{t \in [T]} \{\widehat{\sigma}_t\}$ is the maximal number of outstanding observations after skipping and*

$$S^* = \mathcal{O}\left( \min\left( d_{\max} K^{1/3} \log K, \min_{\mathcal{S} \subseteq [T]} \left\{ |\mathcal{S}| + \sqrt{\mathcal{D}_{\bar{\mathcal{S}}} K^{\frac{2}{3}} \log K} \right\} \right) \right).$$

*Furthermore, if the losses are stochastic, the pseudo-regret also satisfies*

$$\overline{Reg}_T = \mathcal{O}\left( \sum_{i \neq i^*} \left( \frac{\log T}{\Delta_i} + \frac{\widehat{\sigma}_{\max}}{\Delta_i \log K} \right) + K\widehat{\sigma}_{\max} + S^* \right).$$

Masoudian et al. (2022) provide an $\Omega\left(\sqrt{KT} + \min_{\mathcal{S} \subset [T]} \left\{ |\mathcal{S}| + \sqrt{\mathcal{D}_{\bar{\mathcal{S}}} \log K} \right\}\right)$ regret lower bound for adversarial environments with variable delays, which is matched within constants by the algorithm of (Zimmert and Seldin, 2020) for adversarial environments. Our algorithm matches the lower bound within a multiplicative factor of $K^{\frac{1}{3}}$ on the delay-dependent term, which is the price we pay for obtaining a best-of-both-worlds guarantee. The price comes from a reduction of the skipping threshold of Zimmert and Seldin (2020) that we had to make to control the distribution drift that is due to the loss shift (see Appendix B.2). It is an open question whether this factor can be reduced.

In the stochastic regime, assuming that the delays in the first $\sigma_{\max}$ rounds are of order $T$, and that the losses come from Bernoulli distributions with bias close to $\frac{1}{2}$, a trivial regret lower bound is $\Omega\left(\sigma_{\max} \frac{\sum_{i \neq i^*} \Delta_i}{K} + \sum_{i \neq i^*} \frac{\log T}{\Delta_i}\right)$. This bound is almost matched by the algorithm of Joulani et al. (2013) for the stochastic regime only. Our bound has some extra terms, most notably $\sum_{i \neq i^*} \frac{\widehat{\sigma}_{\max}}{\Delta_i \log K}$ and $S^*$. It is an open question whether these terms are inevitable or can be reduced.

Theorem 1 provides three major improvements relative to the results of Masoudian et al. (2022): (1) it requires no advance knowledge of $d_{\max}$; (2) it replaces terms dependent on $d_{\max}$ by terms dependent on $\widehat{\sigma}_{\max}$, which never exceeds $d_{\max}$, and in some cases may be significantly smaller; and (3) it makes skipping possible and beneficial, making the algorithm robust to a small number of excessively large delays and replacing $\sqrt{D \log K}$ term with $\min_{\mathcal{S} \subseteq [T]} \left\{ |\mathcal{S}| + \sqrt{\mathcal{D}_{\bar{\mathcal{S}}} K^{\frac{2}{3}} \log K} \right\}$, which is never much larger, but in some cases significantly smaller.

# 5 Analysis

In this section, we present a proof of Theorem 1. We begin with the stochastic part of the bound in Section 5.1, followed by the adversarial part in Section 5.2.

## 5.1 Stochastic Analysis

We start by defining the drifted regret $\overline{Reg}_T^{drift} = \mathbb{E}\left[ \sum_{t=1}^T \left( \langle x_t, \widehat{\ell}_t^{obs} \rangle - \widehat{\ell}_{t,i_T^*}^{obs} \right) \right]$, where $\widehat{\ell}_t^{obs} = \sum_{s=1}^t \widehat{\ell}_s \mathbb{1}(s + \widehat{d}_s = t)\mathbb{1}(s \notin \mathcal{S}_t)$ is the cumulative vector of losses received at time $t$. Lemma 2 is the first major contribution establishing a relationship between $\overline{Reg}_T^{drift}$ and the actual regret $\overline{Reg}_T$.

**Lemma 2** (Drift of the Drifted Regret). *Let $\sigma_{\max}^t = \max_{s \in [t]} \{\widehat{\sigma}_s\}$. Then*

$$\overline{Reg}_T^{drift} \geq \frac{1}{4}\overline{Reg}_T - 2K \sum_{t=1}^{T} \left( \lambda_{t,t+\widehat{d}_t} + \lambda_{t,t+\widehat{d}_t+\sigma_{\max}^t} \right) - \frac{\sigma_{\max}}{4} - S^*,$$

*where $S^*$ is the total number of rounds skipped by the algorithm.*

The core of Lemma 2 is based on controlling the distribution drift using implicit exploration and skipping. In prior work on bounded delays the relation between $\overline{Reg}_T^{drift}$ and $\overline{Reg}_T$ was achieved by shifting all the arrivals by $d_{\max}$, leading to an additive term of order $d_{\max}$. This approach fails for unbounded delays, because a single delay of order $T$ prevents shifting and leads to linear regret. We address the challenge by introducing a procedure to rearrange the arrivals (Algorithm 2 below) and advanced control of the drift (Lemma 3 below). A proof of Lemma 2 is provided at the end of the section.

---

**Algorithm 2:** Greedy Rearrangement

---
1 **Initialize** $v_t^{new} = 0$ *for all* $t = 1, \ldots, T + d_{\max}^T$
2 **for** $t = 1, \ldots, T$ **do**
3      **for** $s = 1, \ldots, t : s + \widehat{d}_s = t$ **do**
4          Find the first round $\pi(s) \in [t, t + d_{\max}^t]$ such that $v_{\pi(s)}^{new} = 0$
5          Move the arrival from round $s$ to round $\pi(s)$ and update $v_{\pi(s)}^{new} = 1$

---

The drift control lemma (Lemma 3) is the second major contribution of the paper. Prior work on bounded delays controlled the drift by slowing the learning rate in accordance with $d_{\max}$. This does not work for highly varying delays, because slow learning rates prevent learning, whereas fast learning rates fail to control the drift. Lemma 3 relies on implicit exploration terms in the loss estimators in equation (1) and on skipping of excessive delays, leaving the learning rates intact.

**Lemma 3** (Drift Control Lemma). *Let $d_{\max}^t$ be the skipping threshold at time $t$. Then, for any $i \in [K]$ and $s, t \in [T]$, where $s \leq t$ and $t - s \leq d_{\max}^t$, we have*

$$x_{t,i} \leq 4 \max(x_{s,i}, \lambda_{s,t}).$$

The proof is based on introduction of an intermediate variable $\widetilde{x}_s = \nabla \bar{F}_s^*(-\widehat{L}_{t-1}^{obs})$, which is based on the regularizer from round $s$ and the loss estimate from round $t$. It exploits the implicit exploration term $\lambda_{s,t}$ to show that $\frac{x_{t,i}}{\max(\widetilde{x}_i, \lambda_{s,t})} \leq 2$ and skipping to show that $\frac{\widetilde{x}_i}{x_{s,i}} \leq 2$. The latter implies that $\frac{\max(\widetilde{x}_i, \lambda_{s,t})}{\max(x_{s,i}, \lambda_{s,t})} \leq 2$, and in combination with the former completes the proof. The details of the two steps are provided in Appendix B.

Given Lemmas 2 and Lemma 3, we apply standard FTRL analysis, similar to Masoudian et al. (2022), to obtain an upper bound for $\overline{Reg}_T^{drift}$. Specifically, in Appendix A we show that

$$\overline{Reg}_T^{drift} \leq \mathbb{E}\left[ a\sum_{t=1}^{T}\sum_{i \neq i^*} \eta_t x_{t,i}^{1/2} + b\sum_{t=1}^{T}\sum_{i \neq i^*} \gamma_{t+\widehat{d}_t}(v_{t+\widehat{d}_t} - 1)x_{t,i}\Delta_i + c\sum_{t=2}^{T}\sum_{i=1}^{K} \frac{\widehat{\sigma}_t \gamma_t x_{t,i}\log(1/x_{t,i})}{\log K} \right]$$

$$+ \mathcal{O}\left( K\sum_{t=1}^{T} \lambda_{t,t+\widehat{d}_t} \right), \tag{4}$$

where $a, b, c \geq 0$ are constants and $v_t = \sum_{s=1}^{t} \mathbb{1}\left(s + \widehat{d}_s = t\right)$ is the number of arrivals at time $t$ (if a round $s$ is skipped at time $t$ it counts as an "empty" arrival with loss estimate set to zero). By combining (4) with Lemma 2, we obtain

$$\overline{Reg}_T \leq \mathbb{E}\left[ 2a\sum_{t=1}^{T}\sum_{i \neq i^*} \eta_t x_{t,i}^{1/2} + 2b\sum_{t=1}^{T}\sum_{i \neq i^*} \gamma_{t+\widehat{d}_t}(v_{t+\widehat{d}_t} - 1)x_{t,i}\Delta_i + 2c\sum_{t=2}^{T}\sum_{i=1}^{K} \frac{\widehat{\sigma}_t \gamma_t x_{t,i}\log(1/x_{t,i})}{\log K} \right]$$

$$+ \mathcal{O}\left( K\sum_{t=1}^{T} \left( \lambda_{t,t+\widehat{d}_t} + \lambda_{t,t+\widehat{d}_t+\sigma_{\max}^t} \right) + \sigma_{\max} + S^* \right). \tag{5}$$

Then we apply a self-bounding analysis, similar to Masoudian et al. (2022), and get

$$\overline{Reg}_T = \mathcal{O}\left( \sum_{i \neq i^*} \left( \frac{1}{\Delta_i} \log(T) + \frac{\sigma_{\max}}{\Delta_i \log K} \right) + \sigma_{\max} + K \sum_{t=1}^{T} \left( \lambda_{t,t+\widehat{d}_t} + \lambda_{t,t+\widehat{d}_t+\sigma_{\max}^t} \right) + S^* \right).$$

The details of the self-bounding analysis are provided in Appendix C.

The stochastic analysis is completed by the following lemma, which bounds the sum of implicit exploration terms above. It constitutes the third key result of the paper and shows that the bias from implicit exploration does not deteriorate neither the stochastic nor the adversarial bound. The proof is based on a careful study of the evolution of $\mathcal{D}_t$ throughout the game, and is deferred to Appendix D.

**Lemma 4** (Summation Bound). *For all $s \in [T]$, let $\mathcal{D}_s = \sum_{r=1}^{s} \widehat{\sigma}_r$ and $\lambda_{s,t} = e^{-\frac{\mathcal{D}_t}{\mathcal{D}_t - \mathcal{D}_s}}$, then*

$$\sum_{t=1}^{T} \left( \lambda_{t,t+\widehat{d}_t} + \lambda_{t,t+\widehat{d}_t+\sigma_{\max}^t} \right) = \mathcal{O}(\widehat{\sigma}_{\max}).$$

**Proof of Lemma 2 (Drift of the Drifted Regret)**

We start with the definition of the drifted regret.

$$\overline{Reg}_T^{drift} = \mathbb{E}\left[ \sum_{t=1}^{T} \left( \langle x_t, \widehat{\ell}_t^{obs} \rangle - \widehat{\ell}_{t,i_T^*}^{obs} \right) \right] = \sum_{t=1}^{T} \sum_{\substack{s+\widehat{d}_s=t \\ s \notin \mathcal{S}_t}} \sum_{i=1}^{K} \mathbb{E}\left[ \frac{\ell_{s,i} x_{s,i} x_{t,i}}{\max\{x_{s,i}, \lambda_{s,t}\}} - \frac{\ell_{s,i_T^*} x_{s,i_T^*} x_{t,i}}{\max\{x_{s,i_T^*}, \lambda_{s,t}\}} \right]$$

$$\geq \sum_{t=1}^{T} \sum_{\substack{s+\widehat{d}_s=t \\ s \notin \mathcal{S}_t}} \sum_{i=1}^{K} \mathbb{E}\left[ \frac{\ell_{s,i} x_{s,i} x_{t,i}}{\max\{x_{s,i}, \lambda_{s,t}\}} - \ell_{s,i_T^*} x_{t,i} \right]$$

$$\geq \sum_{t=1}^{T} \sum_{s+\widehat{d}_s=t} \sum_{i=1}^{K} \mathbb{E}\Bigg[ \underbrace{\frac{\ell_{s,i} x_{s,i} x_{t,i}}{\max\{x_{s,i}, \lambda_{s,t}\}} - \ell_{s,i_T^*} x_{t,i}}_{\star} \Bigg] - S^*.$$

$$(6)$$

Note that when taking the expectation, we rely on the fact that $\widehat{\ell}_s$ with $s + \widehat{d}_s = t$ does not affect $x_t$. If $\max\{x_{s,i}, \lambda_{s,t}\} = x_{s,i}$, then $\star = \ell_{s,i} x_{t,i}$, otherwise

$$\star = \ell_{s,i} x_{t,i} - \frac{\ell_{s,i} x_{t,i} (\lambda_{s,t} - x_{s,i})}{\lambda_{s,t}} \geq \ell_{s,i} x_{t,i} - \frac{4\lambda_{s,t}(\lambda_{s,t} - x_{s,i})}{\lambda_{s,t}} \geq \ell_{s,i} x_{t,i} - 4\lambda_{s,t}, \quad (7)$$

where the first inequality uses $x_{t,i} \leq 4\max(x_{s,i}, \lambda_{s,t}) = 4\lambda_{s,t}$ by Lemma 3, and $\ell_{s,i} \geq 1$, and the second inequality follows by $x_{s,i} \geq 0$. Plugging (7) into (6) gives

$$\overline{Reg}_T^{drift} \geq \sum_{t=1}^{T} \sum_{s+\widehat{d}_s=t} \sum_{i=1}^{K} \mathbb{E}\left[ \left( \ell_{s,i} x_{t,i} - 4\lambda_{s,t} - \ell_{s,i_T^*} x_{t,i} \right) \right] - S^*$$

$$\geq \mathbb{E}\Bigg[ \underbrace{\sum_{t=1}^{T} \sum_{s+\widehat{d}_s=t} \sum_{i=1}^{K} \Delta_i x_{t,i}}_{R_T} \Bigg] - 4K \sum_{t=1}^{T} \sum_{s+\widehat{d}_s=t} \mathbb{E}[\lambda_{s,t}] - S^*. \quad (8)$$

It suffices to give a lower bound for $R_T$ in terms of the actual regret $\overline{Reg}_T$. The difference between $R_T$ and $\overline{Reg}_T$ is that $\overline{Reg}_T = \mathbb{E}\left[ \sum_{t=1}^{T} \sum_{i=1}^{K} \Delta_i x_{t,i} \right]$, whereas in $R_T$ the sum $\sum_{i=1}^{K} \Delta_i x_{t,i}$ is multiplied by the number of arrivals $\upsilon_t = \sum_{s=1}^{t} \mathbb{1}\left( s + \widehat{d}_s = t \right)$ at time $t$, and $\upsilon_t$ might be larger than one or zero due to delays.

Our main idea here is to leverage the drift control lemma to provide a lower bound for $R_T$ in terms of $\overline{Reg}_T$. Specifically, by Lemma 3 for all $r \in [0, d_{\max}^t]$, we have $\max(x_{t,i}, \lambda_{t,t+r}) \geq \frac{1}{4} x_{t+r,i}$, which implies $x_{t,i} \geq \frac{1}{4} x_{t+r,i} - \lambda_{t,t+r}$. Thus, we obtain the following bound for any $r \in [0, d_{\max}^t]$

$$\sum_{i=1}^{K} \Delta_i x_{t,i} \geq \frac{1}{4} \sum_{i=1}^{K} \Delta_i x_{t+r,i} - K \lambda_{t,t+r}. \tag{9}$$

In Algorithm 2 we provide a greedy procedure to rearrange the arrivals by postponing some arrivals to future rounds to create a *hypothetical* rearranged sequence with at most one arrival at each round. Colliding arrivals are postponed to the first available (unoccupied) slot in the future. In Lemma 5 below we show that arrival originally received at time $t$ stays in the $[t, t + \sigma_{\max}^t]$ interval (note that $\sigma_{\max}^t \leq d_{\max}^t$). When an observation from round $s$ is postponed from arriving at round $t$ to arriving at round $t + r$ for $r \in [0, d_{\max}^t]$, by (9) it is equivalent to replacing $\sum_{i=1}^{K} \Delta_i x_{t,i}$ by $\frac{1}{4} \sum_{i=1}^{K} \Delta_i x_{t+r,i} - K \lambda_{t,t+r}$ in $R_T$. Note that Algorithm 2 may push an arrival to a round larger than $T$, which is equivalent to replacing $\sum_{i=1}^{K} \Delta_i x_{t,i}$ by zero.

Let $v_t^{new}$ for all $t \in [T + d_{\max}^T]$ be the total arrivals at time $t$ after the rearrangement, and let $\pi(t)$ be the round to which we have mapped round $t$ for all $t \in [T]$. Then for any rearrangement

$$R_T = \mathbb{E}\left[ \sum_{t=1}^{T} v_t \sum_{i=1}^{K} \Delta_i x_{t,i} \right] \geq \mathbb{E}\left[ \sum_{t=1}^{T} \frac{1}{4} v_t^{new} \sum_{i=1}^{K} \Delta_i x_{t,i} - K \sum_{t=1}^{T} \lambda_{t,\pi(t)} \right]. \tag{10}$$

The following lemma provides properties of the rearrangement procedure.

**Lemma 5.** *Let $\sigma_{\max}^t = \max_{s \in [t]}\{\widehat{\sigma}_s\}$. Then Algorithm 2 ensures for any $t \in [T + d_{\max}^T]$ that $v_t^{new} \in \{0, 1\}$. Furthermore, for any round $t \in [T]$ it keeps all the arrivals at time $t$ in the interval $[t, t + \sigma_{\max}^t]$, such that $\forall s \leq t : s + \widehat{d}_s = t \Rightarrow \pi(s) - t \leq \sigma_{\max}^t$.*

We provide a proof of the lemma in Appendix E. As a corollary, after the Greedy Rearrangement (Algorithm 2) the number of rounds with zero arrivals is at most $\sigma_{\max}^T$. This is because there will be no arrivals after $T + \sigma_{\max}^T$ and $\sum_{t=1}^{T+\sigma_{\max}^T} v_t^{new} = \sum_{t=1}^{T} v_t = T$, which implies there are at most $\sigma_{\max}^T$ zero arrivals as each round receives at most one arrival. Therefore

$$\mathbb{E}\left[ \sum_{t=1}^{T} v_t^{new} \sum_{i=1}^{K} \Delta_i x_{t,i} \right] = \overline{Reg}_T - \mathbb{E}\left[ \sum_{t=1}^{T} \mathbb{1}(v_t^{new} = 0) \sum_{i=1}^{K} \Delta_i x_{t,i} \right]$$

$$\leq \overline{Reg}_T - \mathbb{E}\left[ \sum_{t=1}^{T} \mathbb{1}(v_t^{new} = 0) \right] \leq \overline{Reg}_T - \mathbb{E}\left[ \sigma_{\max}^T \right] \leq \overline{Reg}_T - \sigma_{\max}, \tag{11}$$

where the first equality uses the definition of $\overline{Reg}_T = \mathbb{E}[\sum_{t=1}^{T} \sum_{i=1}^{K} \Delta_i x_{t,i}]$ and that $\forall t \in [T] : v_t^{new} \in \{0, 1\}$.

Since $\forall t \in [T] : \pi(t) \leq t + \widehat{d}_t + \sigma_{\max}^t$, we have $\lambda_{t,\pi(t)} \leq \lambda_{t,t+\widehat{d}_t+\sigma_{\max}^t}$. Together with (11), (10), and (8) it completes the proof.

## 5.2 Adversarial Analysis

The adversarial analysis is similar to the analysis of Zimmert and Seldin (2020, Theorem 2). In Appendix G we show that

$$\overline{Reg}_T = \mathcal{O}\left( \sqrt{KT} + \min_{\mathcal{S} \subseteq [T]}\left\{ |\mathcal{S}| + \sqrt{\mathcal{D}_{\bar{\mathcal{S}}} \log K} \right\} + S^* + K \sum_{t=1}^{T} \lambda_{t,t+\widehat{d}_t} \right),$$

where the first two terms originate from the analysis of Zimmert and Seldin due to structural similarity of the algorithm, $S^*$ is due to adjusted skipping threshold, and $K \sum_{t=1}^{T} \lambda_{t,t+\widehat{d}_t}$ is due to implicit exploration bias and is bounded by Lemma 4. The proof is completed by the following bound on $S^*$, which is shown in Appendix H.

**Lemma 6.** *We have $S^* = \mathcal{O}\left( \min\left( d_{\max} K^{\frac{2}{3}} \log K , \min_{\mathcal{S} \subseteq [T]}\left\{ |\mathcal{S}| + \sqrt{\mathcal{D}_{\bar{\mathcal{S}}} K^{\frac{2}{3}} \log K} \right\} \right) \right).$*

# 6 Discussion

We have successfully addressed the challenge of handling varying and potentially unbounded delays in best-of-both-worlds setting. The success was based on three technical innovations, which may be interesting in their own right: (1) A relation between the drifted and the standard regret under unbounded delays (given by Lemma 2, Algorithm 2, and Lemma 5); (2) A novel control of distribution drift based on implicit exploration and skipping that does not alter the learning rates and exhibits efficiency under highly varying delays (Lemma 3); and (3) An implicit exploration scheme applicable in best-of-both-worlds setting (Lemma 4).

The work leads to several directions for future research. One question is whether the best-of-both-worlds bounds could be improved further. In particular, whether the $K^{\frac{1}{3}}$ term in the adversarial regret bound could be reduced or eliminated. The term arose due to the need to decrease the skipping threshold of Zimmert and Seldin (2020) to control the distribution drift. It would also be valuable to explore whether it is possible to reduce the $S^*$ term and reduce or eliminate the $\sum_{i \neq i^*} \frac{\widehat{\sigma}_{\max}}{\Delta_i \log K}$ term in the stochastic bound, or to derive lower bounds showing that these terms are unavoidable. Another interesting direction is to find more applications for implicit exploration and skipping in the context of best-of-both-worlds bounds.

## Acknowledgments and Disclosure of Funding

SM and YS acknowledge partial support by the Independent Research Fund Denmark, grant number 9040-00361B.

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

# A  Details of the Drifted Regret Analysis

In this section we prove the bound on drifted regret in equation (4). The derivation is same as the one by Masoudian et al. (2022), however, for the sake of completeness we reproduce it here. The analysis follows the standard FTRL approach, decomposing the drifted pseudo-regret into *penalty* and *stability* terms as

$$\overline{Reg}_T^{drift} = \mathbb{E}\left[\underbrace{\sum_{t=1}^{T}\langle x_t, \widehat{\ell}_t^{obs}\rangle + \bar{F}_t^*(-\widehat{L}_{t+1}^{obs}) - \bar{F}_t^*(-\widehat{L}_t^{obs})}_{stability}\right] + \mathbb{E}\left[\underbrace{\sum_{t=1}^{T}\bar{F}_t^*(-\widehat{L}_t^{obs}) - \bar{F}_t^*(-\widehat{L}_{t+1}^{obs}) - \widehat{\ell}_{t,i_T^*}}_{penalty}\right].$$

The penalty term is bounded by the following inequality, derived by Abernethy et al. (2015)

$$penalty \leq \sum_{t=2}^{T}(F_{t-1}(x_t) - F_t(x_t)) + F_T(e_{i_T^*}) - F_1(x_1), \tag{12}$$

where $e_{i_T^*}$ represents the unit vector in $\mathbb{R}^K$ with the $i_T^*$-th element being one and zero elsewhere. This leads to the following bound for penalty term

$$penalty \leq \mathcal{O}\left(\sum_{t=2}^{T}\sum_{i\neq i^*}\eta_t x_{t,i}^{\frac{1}{2}} + \sum_{t=2}^{T}\sum_{i=1}^{K}\frac{\sigma_t\gamma_t x_{t,i}\log(1/x_{t,i})}{\log K}\right), \tag{13}$$

where we substitute the explicit form of the regularizer into (12) and exploit the properties $\eta_t^{-1} - \eta_{t-1}^{-1} = \mathcal{O}(\eta_t)$, $\gamma_t^{-1} - \gamma_{t-1}^{-1} = \mathcal{O}(\sigma_t\gamma_t/\log K)$, and $x_{t,i_T^*}^{\frac{1}{2}} - 1 \leq 0$.

For the stability term, following a similar analysis as presented by Masoudian et al. (2022, Lemma 5), but incorporating implicit exploration terms, for any $\alpha_t \leq \gamma_t^{-1}$ we obtain

$$stability \leq \sum_{t=1}^{T}\sum_{i=1}^{K}2f_t''(x_{t,i})^{-1}(\widehat{\ell}_{t,i}^{obs} - \alpha_t)^2.$$

Let $A_t = \left\{s \leq t : s + \widehat{d}_s = t\right\}$, then due to the choice of skipping threshold, $\alpha_t = \sum_{s\in A_t}\bar{\ell}_{s,t}$ satisfies the condition $\alpha_t \leq \gamma_t^{-1}$, where $\bar{\ell}_{s,t} = \frac{\sum_{i=1}^{K}f_t''(x_{t,i})^{-1}\widehat{\ell}_{s,i}}{\sum_{i=1}^{K}f_t''(x_{t,i})^{-1}} = \frac{f_t''(x_{t,I_s})^{-1}\widehat{\ell}_{s,I_s}}{\sum_{i=1}^{K}f_t''(x_{t,i})^{-1}}$. Thus we have

$$stability \leq \sum_{t=1}^{T}\sum_{i=1}^{K}2f_t''(x_{t,i})^{-1}\left(\sum_{s\in A_t}\widehat{\ell}_{s,i} - \bar{\ell}_{s,t}\right)^2$$

$$= \underbrace{\sum_{t=1}^{T}\sum_{i=1}^{K}\sum_{s\in A_t}2f_t''(x_{t,i})^{-1}\left(\widehat{\ell}_{s,i} - \bar{\ell}_{s,t}\right)^2}_{S_1}$$

$$+ \underbrace{\sum_{t=1}^{T}\sum_{i=1}^{K}\sum_{r,s\in A_t, r\neq s}2f_t''(x_{t,i})^{-1}\left(\widehat{\ell}_{s,i} - \bar{\ell}_{s,t}\right)\left(\widehat{\ell}_{r,i} - \bar{\ell}_r\right)}_{S_2}$$

For brevity we define $z_{t,i} = f_t''(x_{t,i})^{-1}$ and $m_{s,i}^t = \max\{x_{s,i}, \lambda_{s,t}\}$ for any $s \le t$ and $i \in [K]$. We begin bounding $S_1$ by replacing definition of loss estimators from (1) and get

$$\mathbb{E}[S_1] = \sum_{t=1}^{T} \sum_{i=1}^{K} \sum_{s \in A_t} 2\mathbb{E}\left[z_{t,i}\left(\frac{\ell_{s,I_s}\mathbb{1}(I_s = i)}{m_{s,i}^t} - \frac{z_{t,I_s}\ell_{s,I_s}}{m_{s,I_s}^t \sum_{j=1}^{K} z_{t,j}}\right)^2\right]$$

$$\le \sum_{t=1}^{T} \sum_{i=1}^{K} \sum_{s \in A_t} 2\mathbb{E}\left[z_{t,i}\left(\frac{\mathbb{1}(I_s = i)}{m_{s,i}^t} - \frac{z_{t,I_s}}{m_{s,I_s}^t \sum_{j=1}^{K} z_{t,j}}\right)^2\right]$$

$$= \sum_{t=1}^{T} \sum_{s \in A_t} 2\underbrace{\sum_{i=1}^{K} \mathbb{E}\left[z_{t,i}\left(\frac{\mathbb{1}(I_s = i)}{m_{s,i}^t}^2 - \frac{z_{t,I_s}\mathbb{1}(I_s = i)}{m_{s,i}^t m_{s,I_s}^t \sum_{j=1}^{K} z_{t,j}}\right)\right]}_{S_1^1}$$

$$+ \sum_{t=1}^{T} \sum_{s \in A_t} 2\underbrace{\mathbb{E}\left[\left(\frac{z_{t,I_s}^2}{m_{s,I_s}^t}^2 (\sum_{j=1}^{K} z_{t,j}) - \sum_{i=1}^{K} \frac{z_{t,I_s}z_{t,i}\mathbb{1}(I_s = i)}{m_{s,i}^t m_{s,I_s}^t \sum_{j=1}^{K} z_{t,j}}\right)\right]}_{S_1^2}$$

Where the first inequality uses $\ell_{s,I_s} \le 1$. We show that $S_1^2$ has negative contribution to $S_1$ by taking expectation w.r.t. $I_s$ as the following

$$S_1^2 = \sum_{t=1}^{T} \sum_{s \in A_t} \mathbb{E}\left[\sum_{i=1}^{K} \frac{z_{t,i}^2 x_{s,i}}{m_{s,i}^t}^2 (\sum_{j=1}^{K} z_{t,j}) - \sum_{i=1}^{K} \frac{z_{t,i}^2 x_{s,i}}{m_{s,i}^t}^2 \sum_{j=1}^{K} z_{t,j}\right] = 0$$

Thus we only need to bound $S_1^1$, for which we take expectation w.r.t. $I_s$ and separate $i^*$ from the other arms to get

$$S_1^1 = \sum_{i=1}^{K} \mathbb{E}\left[z_{t,i}\left(\frac{\mathbb{1}(I_s = i)}{m_{s,i}^t}^2 - \frac{z_{t,I_s}\mathbb{1}(I_s = i)}{m_{s,i}^t m_{s,I_s}^t \sum_{j=1}^{K} z_{t,j}}\right)\right]$$

$$\le \sum_{i \ne i^*} \mathbb{E}\left[\frac{z_{t,i} x_{s,i}}{m_{s,i}^t}^2\right] + \mathbb{E}\left[\frac{z_{t,i^*} x_{s,i^*}}{m_{s,i^*}^t}^2 - \frac{z_{t,i^*}^2 x_{s,i^*}}{m_{s,i^*}^t}^2 \sum_{j=1}^{K} z_{t,j}\right]$$

$$\le \sum_{i \ne i^*} \mathbb{E}\left[4\eta_t x_{s,i}^{1/2}\right] + \mathbb{E}\left[\frac{x_{s,i^*}}{m_{s,i^*}^t}^2 \times z_{t,i^*}\left(1 - \frac{z_{t,i^*}}{\sum_{j=1}^{K} z_{t,j}}\right)\right]$$

$$\le \sum_{i \ne i^*} 4\mathbb{E}\left[\eta_t x_{s,i}^{1/2}\right] + \mathbb{E}\left[\frac{x_{s,i^*}}{m_{s,i^*}^t}^2 \times \eta_t x_{t,i^*}^{3/2}\left(1 - \frac{x_{t,i^*}^{3/2}}{(1 - x_{t,i^*})^{3/2} + x_{t,i^*}^{3/2}}\right)\right]$$

$$\le \sum_{i \ne i^*} 4\mathbb{E}\left[\eta_t x_{s,i}^{1/2}\right] + \mathbb{E}\left[\frac{\eta_t x_{s,i^*} x_{t,i^*}^{3/2}}{m_{s,i^*}^t}^2 \times \left(\frac{(1 - x_{t,i^*})^{3/2}}{2^{-1/2}}\right)\right]$$

$$\le \sum_{i \ne i^*} 4\mathbb{E}\left[\eta_t x_{s,i}^{1/2}\right] + \mathbb{E}\left[4\sqrt{2}\eta_t \sum_{i \ne i^*} x_{t,i}\right]$$

$$\le \sum_{i \ne i^*} 4\mathbb{E}\left[\eta_t x_{s,i}^{1/2}\right] + \mathbb{E}\left[16\sqrt{2}\eta_t \sum_{i \ne i^*} (x_{s,i} + \lambda_{s,t})\right]$$

$$\le \mathcal{O}\left(\mathbb{E}\left[\eta_s \sum_{i \ne i^*} x_{s,i}^{1/2}\right] + \mathbb{E}[K\lambda_{s,t}]\right),$$

where the second inequality uses $z_{t,i} = f_t''(x_{t,i})^{-1} \le \eta_t x_{t,i}^{3/2}$ along $x_{t,i} \le m_{s,i}^t$ from Lemma 3, the third inequality is due the fact that $z_{t,i^*}\left(1 - \frac{z_{t,i^*}}{\sum_{j=1}^{K} z_{t,j}}\right)$ is an increasing function in terms

of both $z_{t,i^*}$ and $\sum_{i\neq i^*} z_{t,i}$ and we substitute $z_{t,i^*} \leq \eta_t x_{t,i^*}^{3/2}$ and $\sum_{j\neq i^*} z_{t,j} \leq \sum_{j\neq i^*} \eta_t x_{t,j}^{3/2} \leq$ $\eta_t(1 - x_{t,i^*})^{3/2}$, the fourth inequality is due to $(1-a)^{3/2} + a^{3/2} \leq 2^{-1/2}$, the fifth and the sixth inequalities rely on Lemma 3, and finally the last inequality is followed by $\forall i : x_{s,i} \leq x_{s,i}^{1/2}$ and that $\eta_t \leq \eta_s$. Combining bounds for $S_1^1$ and $S_1^2$ gives the following bound for $S_1$

$$\mathbb{E}[S_1] \leq \mathcal{O}\left(\sum_{t=1}^{T}\sum_{i\neq i^*} \eta_t \mathbb{E}[x_{t,i}^{1/2}] + \sum_{t=1}^{T} K\lambda_{t,t+\widehat{d}_t}\right) \tag{14}$$

For $S_2$, we take expectation with respect to $I_s$, $I_r$, and randomness of losses, all separately to get

$$\mathbb{E}[S_2] = \sum_{t=1}^{T}\sum_{i=1}^{K}\sum_{r,s\in A_t, r\neq s} 2\mathbb{E}\left[z_{t,i}\left(\widehat{\ell}_{s,i} - \bar{\ell}_s\right)\left(\widehat{\ell}_{r,i} - \bar{\ell}_s\right)\right]$$

$$= \sum_{t=1}^{T}\sum_{i=1}^{K}\sum_{r,s\in A_t, r\neq s} 2\mathbb{E}\left[z_{t,i}\left(\frac{\mu_i x_{s,i}}{m_{s,i}^t} - \frac{\sum_{j=1}^{K} z_{t,j}\mu_j x_{s,j}/m_{s,j}^t}{\sum_{j=1}^{K} z_{t,j}}\right)\left(\frac{\mu_i x_{r,i}}{m_{r,i}^t} - \frac{\sum_{j=1}^{K} z_{t,j}\mu_j x_{r,j}/m_{r,j}^t}{\sum_{j=1}^{K} z_{t,j}}\right)\right]. \tag{15}$$

For simplicity we define $\epsilon_{s,i}^t = \mu_i - \frac{\mu_i x_{s,i}}{m_{s,i}^t}$ for any $s \leq t$ and any $i \in [K]$, for which we have the following bounds

$$0 \leq \epsilon_{s,i}^t \leq \frac{\lambda_{s,t}}{m_{s,i}^t}.$$

We then continue from 15 and bound it as the following

$$\mathbb{E}[S_2]$$

$$= \sum_{t=1}^{T}\sum_{\substack{r\neq s \\ r,s\in A_t}}\sum_{i=1}^{K} 2\mathbb{E}\left[z_{t,i}\left(\mu_i - \frac{\sum_{j=1}^{K} z_{t,j}\mu_j}{\sum_{j=1}^{K} z_{t,j}} - \epsilon_{s,i}^t + \frac{\sum_{j=1}^{K} z_{t,j}\epsilon_{s,j}^t}{\sum_{j=1}^{K} z_{t,j}}\right)\left(\mu_i - \frac{\sum_{j=1}^{K} z_{t,j}\mu_j}{\sum_{j=1}^{K} z_{t,j}} - \epsilon_{r,i}^t + \frac{\sum_{j=1}^{K} z_{t,j}\epsilon_{r,j}^t}{\sum_{j=1}^{K} z_{t,j}}\right)\right]$$

$$\leq \sum_{t=1}^{T}\sum_{\substack{r\neq s \\ r,s\in A_t}} 2\mathbb{E}\left[\underbrace{\sum_{i=1}^{K} z_{t,i}\left(\mu_i - \frac{\sum_{j=1}^{K} z_{t,j}\mu_j}{\sum_{j=1}^{K} z_{t,j}}\right)^2}_{S_2^1} + \underbrace{\sum_{i=1}^{K} z_{t,i}\epsilon_{s,i}^t\epsilon_{r,i}^t + 2z_{t,i}(\epsilon_{s,i}^t + \epsilon_{r,i}^t)}_{S_2^2} + \underbrace{\frac{(\sum_{i=1}^{K} z_{t,i}\epsilon_{s,i}^t)(\sum_{i=1}^{K} z_{t,i}\epsilon_{r,i}^t)}{\sum_{i=1}^{K} z_{t,i}}}_{S_2^3}\right], \tag{16}$$

where the inequality holds because we ignore the negative terms after multiplication and that $|(\mu_i - \frac{\sum_{j=1}^{K} z_{t,j}\mu_j}{\sum_{j=1}^{K} z_{t,j}})| \leq 1$. We need to bound each part from (16). We start with $S_2^1$,

$$S_2^1 = \sum_{i=1}^{K} z_{t,i}\left(\mu_i - \frac{\sum_{j=1}^{K} z_{t,j}\mu_j}{\sum_{j=1}^{K} z_{t,j}}\right)^2$$

$$= \sum_{i=1}^{K} z_{t,i}\mu_i^2 - \frac{\left(\sum_{i=1}^{K} z_{t,i}\mu_i\right)^2}{\sum_{i=1}^{K} z_{t,i}}$$

$$\leq \sum_{i=1}^{K} z_{t,i}\mu_i^2 - \frac{\left(\sum_{i=1}^{K} z_{t,i}\mu_{i^*}\right)^2}{\sum_{i=1}^{K} z_{t,i}}$$

$$\leq \sum_{i=1}^{K} z_{t,i}(\mu_i^2 - \mu_{i^*}^2)$$

$$\leq \sum_{i\neq i^*} 2\gamma_t x_{t,i}\Delta_i \tag{17}$$

We bound $S_2^2$ as

$$S_2^2 = \sum_{i=1}^{K} z_{t,i} \epsilon_{s,i}^t \epsilon_{r,i}^t + 2z_{t,i}(\epsilon_{s,i}^t + \epsilon_{r,i}^t)$$

$$\leq \sum_{i=1}^{K} z_{t,i} \frac{\epsilon_{s,i}^t + \epsilon_{r,i}^t}{2} + 2z_{t,i}(\epsilon_{s,i}^t + \epsilon_{r,i}^t)$$

$$\leq \frac{5}{2} \sum_{i=1}^{K} \frac{z_{t,i}\lambda_{s,t}}{m_{s,i}^t} + \frac{z_{t,i}\lambda_{r,t}}{m_{r,i}^t}$$

$$\leq \frac{5}{2} K\gamma_t(\lambda_{s,t} + \lambda_{r,t}), \tag{18}$$

where the last inequality holds because $z_{t,i} \leq \gamma_t x_{t,i}$ and that $x_{t,i} \leq 4m_{s,i}^t, 4m_{r,i}^t$ from Lemma 3. It remains to give upper bound for $S_2^3$ as

$$S_2^3 = \frac{(\sum_{i=1}^{K} z_{t,i}\epsilon_{s,i}^t)(\sum_{i=1}^{K} z_{t,i}\epsilon_{r,i}^t)}{\sum_{i=1}^{K} z_{t,i}}$$

$$\leq \frac{(\sum_{i=1}^{K} z_{t,i}\lambda_{s,t}/m_{s,i}^t)(\sum_{i=1}^{K} z_{t,i}\lambda_{r,t}/m_{r,i}^t)}{\sum_{i=1}^{K} z_{t,i}}$$

$$\leq \frac{1}{2} K\gamma_t(\lambda_{s,t} + \lambda_{r,t}), \tag{19}$$

where the second inequality rely on $z_{t,i} \leq \gamma_t x_{t,i}$, $\lambda_{s,t} \leq m_{s,i}^t$, $\lambda_{r,t} \leq m_{r,i}^t$, and $x_{t,i} \leq 4m_{s,i}^t$, $x_{t,i} \leq 4m_{r,i}^t$ from Lemma 3. It is suffices to plug bounds in (17), (18), and (19) to obtain

$$\mathbb{E}[S_2] \leq \sum_{t=1}^{T} \sum_{i \neq i^*} 4\Delta_i \gamma_t \mathbb{E}[x_{t,i}]\upsilon_t(\upsilon_t - 1) + 6\sum_{t=1}^{T} K\gamma_{t+\widehat{d}_t}(\upsilon_{t+\widehat{d}_t} - 1)\lambda_{t,t+\widehat{d}_t}$$

$$\leq \sum_{t=1}^{T} \sum_{i \neq i^*} \sum_{s \in A_t} 4\Delta_i \gamma_t \mathbb{E}[x_{s,i} + \lambda_{s,t}](\upsilon_t - 1) + 6\sum_{t=1}^{T} K\gamma_{t+\widehat{d}_t}(\upsilon_{t+\widehat{d}_t} - 1)\lambda_{t,t+\widehat{d}_t}$$

$$\leq \sum_{t=1}^{T} \sum_{i \neq i^*} \sum_{s \in A_t} 4\Delta_i \gamma_t \mathbb{E}[x_{s,i}](\upsilon_t - 1) + 10\sum_{t=1}^{T} K\gamma_{t+\widehat{d}_t}(\upsilon_{t+\widehat{d}_t} - 1)\lambda_{t,t+\widehat{d}_t}$$

$$\leq \mathcal{O}\left(\sum_{t=1}^{T} \sum_{i \neq i^*} \gamma_{t+\widehat{d}_t}\Delta_i \mathbb{E}[x_{t,i}](\upsilon_{t+\widehat{d}_t} - 1) + K\sum_{t=1}^{T} \lambda_{t,t+\widehat{d}_t}\right), \tag{20}$$

where the third inequality uses Lemma 3 and the last inequality holds because of the skipping that ensures $\gamma_{t+\widehat{d}_t}(\upsilon_{t+\widehat{d}_t} - 1) \leq 1$. Now, it is sufficient to combine the bounds for $S_1$ and $S_2$ in (14) and (20) and get

$$\mathbb{E}[stability] \leq \mathcal{O}\left(\sum_{t=1}^{T} \sum_{i \neq i^*} \eta_t \mathbb{E}[x_{t,i}^{1/2}] + \sum_{t=1}^{T} \sum_{i \neq i^*} \gamma_{t+\widehat{d}_t}\mathbb{E}[x_{t,i}](\upsilon_{t+\widehat{d}_t} - 1) + K\sum_{t=1}^{T} \lambda_{t,t+\widehat{d}_t}\right). \tag{21}$$

Combining the stability bound from (21) and the penalty bound from (13) concludes the proof.

## B    Proof of the Drift Control Lemma

In this section we provide a proof of Lemma 3. We start with a few auxiliary results, and then prove the lemma.

### B.1    Auxiliary results for the proof of the key lemma

For the proof we use two facts and a lemma from Masoudian et al. (2022), and a new lemma. Recall that $f_t(x) = -2\eta_t^{-1}\sqrt{x} + \gamma_t^{-1}x\log x$.

**Fact 7.** *(Masoudian et al., 2022, Fact 15)* $f_t'(x)$ *is a concave function.*

**Fact 8.** *(Masoudian et al., 2022, Fact 16)* $f_t''(x)^{-1}$ *is a convex function.*

**Lemma 9.** *(Masoudian et al., 2022, Lemma 17) Fix $t$ and $s$ with $t \geq s$, and assume that there exists $\alpha$, such that $x_{t,i} \leq \alpha \max(x_{s,i}, \lambda_{s,t})$ for all $i \in [K]$, and let $f(x) = \left(-2\eta_t^{-1}\sqrt{x} + \gamma_t^{-1}x \log x\right)$, then we have the following inequality*

$$\frac{\sum_{j=1}^{K} f''(x_{t,j})^{-1}\widehat{\ell}_{s,j}}{\sum_{j=1}^{K} f''(x_{t,j})^{-1}} \leq 2\alpha(K-1)^{\frac{1}{3}}.$$

**Lemma 10.** *If $t > s$ and $(t-s) \leq d_{\max}^t$, then*

$$d_{\max}^t \leq \sqrt{2}d_{\max}^s,$$

*which is equivalent to $\mathcal{D}_t \leq 2\mathcal{D}_s$.*

*Proof.* It suffices to prove that $\mathcal{D}_t \leq 2\mathcal{D}_s$, which is equivalent to proving that $(\mathcal{D}_t - \mathcal{D}_s) \leq \frac{1}{2}\mathcal{D}_t$. We have:

$$\mathcal{D}_t - \mathcal{D}_s = \sum_{r=s+1}^{t} \widehat{\sigma}_r \leq (t-s)d_{\max}^t \leq \left(d_{\max}^t\right)^2 = \frac{\mathcal{D}_t}{49K^{\frac{2}{3}}\log K} \leq \frac{\mathcal{D}_t}{2},$$

where the first inequality holds because due to skipping, for all $r \leq t$ we have $\widehat{\sigma}_r \leq d_{\max}^t$, and $(t-s) \leq d_{\max}^t$. ∎

### B.2 Proof of the Drift Control Lemma

Now we are ready to provide a proof of Lemma 3. Similar to the analysis of Masoudian et al. (2022), the proof relies on induction on *valid* pairs $(t,s)$, where a pair $(t,s)$ is considered valid if $s \leq t$ and $(t-s) \leq d_{\max}^t$. The induction step for pair $(t,s)$ involves proving that $x_{t,i} \leq 4\max(x_{s,i}, \lambda_{s,t})$ for all $i \in [K]$. To establish this, we use the induction assumption for all valid pairs $(t',s')$ such that $s', t' < t$, as well as all valid pairs $(t',s')$, such that $t' = t$ and $s < s' \leq t$. The induction base encompasses all pairs $(t', t')$ for all $t' \in [T]$, where the statement $x_{t',i} \leq 4x_{t',i}$ holds trivially.

To control $\frac{x_{t,i}}{\max(x_{s,i}, \lambda_{s,t})}$ we first introduce an auxiliary variable $\widetilde{x} = \bar{F}_s^*(-\widehat{L}_{t-1}^{obs})$. We then address the problem of drift control by breaking it down into two sub-problems:

1. $\frac{x_{t,i}}{\max(\widetilde{x}_i, \lambda_{s,t})} \leq 2$: the drift due to change of regularizer,

2. $\frac{\widetilde{x}_i}{x_{s,i}} \leq 2$: the drift due to loss shift.

**Deviation induced by the change of regularizer**

The regularizer at round $r$ is defined as

$$F_r(x) = \sum_{i=1}^{K} f_r(x_i) = \sum_{i=1}^{K} \left(-2\eta_r^{-1}\sqrt{x_i} + \gamma_r^{-1}x_i \log x_i\right).$$

We have $x_t = \nabla \bar{F}_t^*(-\widehat{L}_{t-1}^{obs})$ and $\widetilde{x} = \nabla \bar{F}_s^*(-\widehat{L}_{t-1}^{obs})$. According to the KKT conditions, there exist Lagrange multipliers $\mu$ and $\widetilde{\mu}$, such that for all $i$:

$$f_s'(\widetilde{x}_i) = -\widehat{L}_{t-1,i}^{obs} + \widetilde{\mu},$$
$$f_t'(x_{t,i}) = -\widehat{L}_{t-1,i}^{obs} + \mu.$$

We also know that there exists an index $j$, such that $\widetilde{x}_j \geq x_{t,j}$. This leads to the following inequality:

$$-\widehat{L}_{t-1,j}^{obs} + \mu = f_t'(x_{t,j}) \leq f_s'(x_{t,j}) \leq f_s'(\widetilde{x}_j) = -\widehat{L}_{t-1,j}^{obs} + \widetilde{\mu},$$

where the first inequality holds because the learning rates are decreasing, and the second inequality is due to the fact that $f'_s(x)$ is increasing. This implies that $\mu \leq \widetilde{\mu}$, which gives us the following inequality for all $i$:

$$f'_t(x_{t,i}) = -\frac{1}{\eta_t\sqrt{x_{t,i}}} + \frac{\log(x_{t,i})}{\gamma_t} \leq -\frac{1}{\eta_s\sqrt{\widetilde{x}_i}} + \frac{\log(\widetilde{x}_i)}{\gamma_s} = f'_s(\widetilde{x}_i).$$

Thus, we have two cases, either $-\frac{1}{\eta_t\sqrt{x_{t,i}}} \leq -\frac{1}{\eta_s\sqrt{\widetilde{x}_i}}$ or $\frac{\log(x_{t,i})}{\gamma_t} \leq \frac{\log(\widetilde{x}_i)}{\gamma_s}$.

**Case i:** If $-\frac{1}{\eta_t\sqrt{x_{t,i}}} \leq -\frac{1}{\eta_s\sqrt{\widetilde{x}_i}}$ holds, then we have $\frac{x_{t,i}}{\widetilde{x}_i} \leq \frac{\eta_s^2}{\eta_t^2} = \frac{t}{s}$. On the other hand, we have

$$t - s \leq d^t_{\max} = \sqrt{\frac{\sum_{r=1}^{t}\widehat{\sigma}_r}{K^{3/2}\log K}} \leq \sqrt{\frac{t^2/2}{K^{3/2}\log K}} \leq \frac{t}{2},$$

where the second inequality holds because trivially $\widehat{\sigma}_r \leq r$. This implies that $\frac{x_{t,i}}{\widetilde{x}_i} \leq 2$.

**Case ii:** If $\frac{\log(x_{t,i})}{\gamma_t} \leq \frac{\log(\widetilde{x}_i)}{\gamma_s}$, it implies that $x_{t,i} \leq \widetilde{x}_i^{\frac{\gamma_t}{\gamma_s}}$. Using $\widetilde{x}_i \leq \max(\widetilde{x}_i, \lambda_{s,t})$, we get

$$
\begin{aligned}
x_{t,i} &\leq \max(\widetilde{x}_i, \lambda_{s,t})^{\frac{\gamma_t}{\gamma_s}}\\
&= \max(\widetilde{x}_i, \lambda_{s,t}) \times \max(\widetilde{x}_i, \lambda_{s,t})^{\frac{\gamma_t}{\gamma_s}-1}\\
&\leq \max(\widetilde{x}_i, \lambda_{s,t}) \times \lambda_{s,t}^{\frac{\gamma_t}{\gamma_s}-1}\\
&= \max(\widetilde{x}_i, \lambda_{s,t}) \times \lambda_{s,t}^{-\frac{\sqrt{\mathcal{D}_t}-\sqrt{\mathcal{D}_s}}{\sqrt{\mathcal{D}_t}}}\\
&= \max(\widetilde{x}_i, \lambda_{s,t}) \times e^{\frac{\mathcal{D}_t}{\mathcal{D}_t-\mathcal{D}_s} \times \frac{\sqrt{\mathcal{D}_t}-\sqrt{\mathcal{D}_s}}{\sqrt{\mathcal{D}_t}}}\\
&= \max(\widetilde{x}_i, \lambda_{s,t}) \times e^{\frac{\sqrt{\mathcal{D}_t}}{(\sqrt{\mathcal{D}_t}+\sqrt{\mathcal{D}_s})}} \leq \max(\widetilde{x}_i, \lambda_{s,t}) \times e^{\frac{1}{1+\sqrt{\frac{1}{2}}}} \leq \max(\widetilde{x}_i, \lambda_{s,t}) \times 2.
\end{aligned}
$$

Therefore, in both cases we obtain

$$x_{t,i} \leq 2\max(\widetilde{x}_i, \lambda_{s,t}). \tag{22}$$

**Deviation Induced by the Loss Shift**

The initial steps of the proof of this part are the same as in Masoudian et al. (2022). However, for the sake of completeness, we restate them here.

Since we have $x_s = \nabla\bar{F}^*_s(-\widehat{L}^{obs}_{s-1})$ and $\widetilde{x} = \nabla\bar{F}^*_s(-\widehat{L}^{obs}_{t-1})$, they both share the same regularizer $F_s(x) = \sum_{i=1}^{K} f_s(x_i)$. For brevity, we drop $s$ from $f_s(x)$. By the KKT conditions $\exists \mu, \widetilde{\mu}$ s.t. $\forall i$:

$$
\begin{aligned}
f'(x_{s,i}) &= -\widehat{L}^{obs}_{s-1,i} + \mu,\\
f'(\widetilde{x}_i) &= -\widehat{L}^{obs}_{t-1,i} + \widetilde{\mu}.
\end{aligned}
$$

Let $\widetilde{\ell} = \widehat{L}^{obs}_{t-1} - \widehat{L}^{obs}_{s-1}$, then by the concavity of $f'(x)$ from Fact 7, we have

$$(x_{s,i} - \widetilde{x}_i)f''(x_{s,i}) \leq \underbrace{f'(x_{s,i}) - f'(\widetilde{x}_i)}_{\mu - \widetilde{\mu} + \widetilde{\ell}_i} \leq (x_{s,i} - \widetilde{x}_i)f''(\widetilde{x}_i). \tag{23}$$

Since $f''(x_{s,i}) \geq 0$, from the left side of (23) we get $x_{s,i} - \widetilde{x}_i \leq f''(x_{s,i})^{-1}\left(\mu - \widetilde{\mu} + \widetilde{\ell}_i\right)$. Taking summation over all $i$ and using the fact that both vectors $x_s$ and $\widetilde{x}$ are probability vectors, we have

$$0 = \sum_{i=1}^{K}(x_{s,i} - \widetilde{x}_i) \leq \sum_{i=1}^{K}f''(x_{s,i})^{-1}\left(\mu - \widetilde{\mu} + \widetilde{\ell}_i\right),$$

$$\Rightarrow \widetilde{\mu} - \mu \leq \frac{\sum_{i=1}^{K}f''(x_{s,i})^{-1}\widetilde{\ell}_i}{\sum_{i=1}^{K}f''(x_{s,i})^{-1}}. \tag{24}$$

Combining the right hand sides of (23) and (24) gives

$$(\widetilde{x}_i - x_{s,i})f''(\widetilde{x}_i) \leq \widetilde{\mu} - \mu - \widetilde{\ell}_i \leq \frac{\sum_{j=1}^{K} f''(x_{s,j})^{-1}\widetilde{\ell}_j}{\sum_{j=1}^{K} f''(x_{s,j})^{-1}},$$

and by rearrangement we get

$$\widetilde{x}_i \leq x_{s,i} + f''(\widetilde{x}_i)^{-1} \times \frac{\sum_{j=1}^{K} f''(x_{s,j})^{-1}\widetilde{\ell}_j}{\sum_{j=1}^{K} f''(x_{s,j})^{-1}}$$

$$\leq x_{s,i} + \gamma_s \widetilde{x}_i \times \frac{\sum_{j=1}^{K} f''(x_{s,j})^{-1}\widetilde{\ell}_j}{\sum_{j=1}^{K} f''(x_{s,j})^{-1}}, \tag{25}$$

where the last inequality holds because $f''(\widetilde{x}_i)^{-1} = \left(\eta_s^{-1}\frac{1}{2}\widetilde{x}_i^{-3/2} + \gamma_s^{-1}\widetilde{x}_i^{-1}\right)^{-1}$. The next step for bounding $\widetilde{x}_i$ is to bound $\frac{\sum_{j=1}^{K} f''(x_{s,j})^{-1}\widetilde{\ell}_j}{\sum_{j=1}^{K} f''(x_{s,j})^{-1}}$ in (25), where $\widetilde{\ell}_j = \sum_{r \in A} \widehat{\ell}_{r,j}$ and $A = \left\{r : s \leq r + \widehat{d}_r < t\right\}$.

If there exists $r \in A$, such that $r > s$ and $4\max(x_{r,i}, \lambda_{r,r+\widehat{d}_r}) \leq x_{s,i}$, then combining it with the induction assumption for $(r + \widehat{d}_r, r)$, where we have $x_{r+\widehat{d}_r,i} \leq 4\max(x_{r,i}, \lambda_{r,r+\widehat{d}_r})$, leads to $x_{r+\widehat{d}_r,i} \leq x_{s,i}$. On the other hand, by the induction assumption for pair $(r + \widehat{d}_r, t)$, we have

$$x_{t,i} \leq 4\max(x_{r+\widehat{d}_r,i}, \lambda_{r+\widehat{d}_r,t}).$$

So using $x_{r+\widehat{d}_r,i} \leq x_{s,i}$ and $\lambda_{r+\widehat{d}_r,t} \leq \lambda_{s,t}$ we can derive $x_{t,i} \leq 4\max(x_{s,i}, \lambda_{s,t})$. This inequality satisfies the condition we wanted to prove in the drift lemma. Therefore, we assume that for all $r \in A$ we have either $r \leq s$ or $x_{s,i} \leq 4\max(x_{r,i}, \lambda_{r,r+\widehat{d}_r})$. If $r \leq s$, using the the induction assumption for $(s, r)$ together with the fact that $\lambda_{r,s} \leq \lambda_{r,r+\widehat{d}_r}$, results in $x_{s,i} \leq 4\max(x_{r,i}, \lambda_{r,s})$. Consequently, in either case, the following inequality holds for all $r \in A$

$$x_{s,i} \leq 4\max(x_{r,i}, \lambda_{r,r+\widehat{d}_r}). \tag{26}$$

Thus, inequality in (26) satisfies the condition of Lemma 9, and for all $r \in A$ we get:

$$\frac{\sum_{j=1}^{K} f''(x_{s,j})^{-1}\widehat{\ell}_{r,j}}{\sum_{j=1}^{K} f''(x_{s,j})^{-1}} \leq 8(K-1)^{\frac{1}{3}}. \tag{27}$$

We proceed by summing both sides of the inequality (27) over all $r \in A$ and obtain $\frac{\sum_{j=1}^{K} f''(x_{s,j})^{-1}\widetilde{\ell}_j}{\sum_{j=1}^{K} f''(x_{s,j})^{-1}} \leq 4|A|(K-1)^{\frac{1}{3}}$. Now it suffices to plug this result into (25):

$$\widetilde{x}_i \leq x_{s,i} + 8|A|\gamma_s \widetilde{x}_i (K-1)^{\frac{1}{3}} \Rightarrow$$

$$\widetilde{x}_i \leq x_{s,i} \times \left(\frac{1}{1 - 8|A|\gamma_s(K-1)^{1/3}}\right) \tag{28}$$

$$\leq x_{s,i} \times \left(\frac{1}{1 - 24\gamma_s d_{\max}^s(K-1)^{1/3}}\right)$$

$$\leq x_{s,i} \times \left(\frac{1}{1 - 1/2}\right) = 2x_{s,i}, \tag{29}$$

where the third inequality uses $|A| \leq d_{\max}^s + t - s \leq d_{\max}^t + d_{\max}^s$, and that $d_{\max}^t \leq 2d_{\max}^s$ by Lemma 10, and for the last inequality we use the definitions of $\gamma_s$ and $d_{\max}^s$.

Combining (29) and (22) completes the induction step.

## C Self-Bounding Analysis

In this section we show the details of how to apply self-bounding analysis to bound the right hand side of (5).

We start from (5) and decompose it as follows

$$
\overline{Reg}_T \leq \mathbb{E}\left[ a \underbrace{\sum_{t=1}^{T}\sum_{i\neq i^*} \eta_t x_{t,i}^{1/2}}_{A} + b \underbrace{\sum_{t=1}^{T}\sum_{i\neq i^*} \gamma_{t+d_t}(v_{t+d_t}-1)x_{t,i}\Delta_i}_{B} + c \underbrace{\sum_{t=2}^{T}\sum_{i=1}^{K} \frac{\widehat{\sigma}_t \gamma_t x_{t,i}\log(1/x_{t,i})}{\log K}}_{C} \right]
$$

$$
+ \underbrace{\mathcal{O}\left( K \sum_{t=1}^{T}\left(\lambda_{t,t+\widehat{d}_t} + \lambda_{t,t+\widehat{d}_t+\sigma_{\max}^t}\right) + \sigma_{\max} + S^* \right)}_{D}.
$$

We rewrite the pseudo-regret as $\overline{Reg}_T = 4\overline{Reg}_T - 3\overline{Reg}_T$, and then based on the decomposition above we have

$$
\overline{Reg}_T \leq \mathbb{E}\left[4aA - \overline{Reg}_T\right] + \mathbb{E}\left[4bB - \overline{Reg}_T\right] + \mathbb{E}\left[4cC - \overline{Reg}_T\right] + 4D. \tag{30}
$$

Masoudian et al. (2022) provide the following three lemmas that give the bounds for the first three terms in (30). Although the algorithm of Masoudian et al. differs from ours, their bounds remain applicable, because they are based on the worst-case choice of $x_{t,i}$, which is algorithm-independent.

**Lemma 11.** *(Masoudian et al., 2022, Lemma 6) For any $a \geq 0$, we have:*

$$
4aA - \overline{Reg}_T \leq \sum_{i\neq i^*} \frac{4a^2}{\Delta_i}\log(T+1) + 1. \tag{31}
$$

**Lemma 12.** *(Masoudian et al., 2022, Lemma 7) Let $v_{max} = \max_{t\in[T]} v_t$, then for any $b \geq 0$:*

$$
4bB - \overline{Reg}_T \leq 64b^2 v_{max}\log K. \tag{32}
$$

It is evident that $v_{max} \leq \sigma_{\max}$, so the bound in Lemma 12 is dominated by $\mathcal{O}(K\sigma_{\max})$ term in the regret bound.

**Lemma 13.** *(Masoudian et al., 2022, Lemma 8) For any $c \geq 0$:*

$$
4cC - \overline{Reg}_T \leq \sum_{i\neq i^*} \frac{128c^2\sigma_{\max}}{\Delta_i \log K}. \tag{33}
$$

By plugging (31),(32),(33) into (30) we get the desired bound.

## D A Proof of Lemma 4

First we provide two facts and two auxiliary lemmas.

**Lemma 14.** *For any $t$ we have*

$$
2\mathcal{D}_t \geq \sum_{s=1}^{t} \widehat{d}_s.
$$

*Proof.* We show that for any $t \in [T]$ we have $\sum_{s=1}^{t} \widehat{d}_s - \mathcal{D}_t \leq \mathcal{D}_t$:

$$
\begin{aligned}
\sum_{s=1}^{t} \widehat{d}_s - \mathcal{D}_t &= \sum_{(s\leq t)\wedge(s+\widehat{d}_s>t)} (\widehat{d}_s - \widehat{\sigma}_s) \\
&\leq \sum_{(s\leq t)\wedge(s+\widehat{d}_s>t)} \widehat{d}_s \\
&\leq \left(d_{\max}^t\right)^2 = \frac{\mathcal{D}_t}{49K^{\frac{2}{3}}\log K} \leq \mathcal{D}_t,
\end{aligned}
$$

where the second inequality holds because $\widehat{d}_s \leq d_{\max}^t$, and the total number of steps that satisfy $(s \leq t) \wedge (s + \widehat{d}_s > t)$ is less than the skipping threshold at time $t$, which is again $d_{\max}^t$. Rearranging the inequality completes the proof. ∎

**Lemma 15** ((Orabona, 2022, Lemma 4.13)). *Let $a_0 \geq 0$ and $f : [0; +\infty) \to [0; +\infty)$ be a nonincreasing function. Then*

$$\sum_{t=1}^{T} a_t f\left(a_0 + \sum_{i=1}^{t} a_i\right) \leq \int_{a_0}^{\sum_{t=0}^{T} a_t} f(x)dx.$$

**Fact 16.** *For any $x \geq 0$, we have $e^{-x} \leq \frac{1}{x}$.*

**Fact 17.** *For any $x \geq 1$, we have $e^{-x} \leq \frac{1}{x \log^2(x)}$.*

*Proof of Lemma 4.* We have two summations as

$$\sum_{t=1}^{T} e^{-\frac{\mathcal{D}_{t+\widehat{d}_t}}{\mathcal{D}_{t+\widehat{d}_t} - \mathcal{D}_t}} + \sum_{t=1}^{T} e^{-\frac{\mathcal{D}_{t+\sigma_{\max}^t + \widehat{d}_t}}{\mathcal{D}_{t+\sigma_{\max}^t + \widehat{d}_t} - \mathcal{D}_t}},$$

where we show an upper bound of $\mathcal{O}(\widehat{\sigma}_{\max})$ for each of them.

**Bounding the First Summation:** Let $T_0$ be the time satisfying $\sqrt{\mathcal{D}_{T_0}} = \frac{\widehat{\sigma}_{\max}}{K^{1/3} \log(K)}$, then using Facts 16 and 17 we have

$$\sum_{t=1}^{T} e^{-\frac{\mathcal{D}_{t+\widehat{d}_t}}{\mathcal{D}_{t+\widehat{d}_t} - \mathcal{D}_t}} \leq \underbrace{\sum_{t=1}^{T_0} \frac{\mathcal{D}_{t+\widehat{d}_t} - \mathcal{D}_t}{\mathcal{D}_{t+\widehat{d}_t}}}_{A} + \underbrace{\sum_{t=T_0+1}^{T} \frac{\mathcal{D}_{t+\widehat{d}_t} - \mathcal{D}_t}{\mathcal{D}_{t+\widehat{d}_t} \log^2\left(\frac{\mathcal{D}_{t+\widehat{d}_t}}{\mathcal{D}_{t+\widehat{d}_t} - \mathcal{D}_t}\right)}}_{B}.$$

For $A$ we give the following bound

$$\begin{aligned}
A = \sum_{t=1}^{T_0} \sum_{s=t+1}^{t+\widehat{d}_t} \frac{\widehat{\sigma}_s}{\mathcal{D}_{t+\widehat{d}_t}} &= \sum_{s=1}^{T_0} \sum_{t=0}^{s-1} \frac{\widehat{\sigma}_s \mathbb{1}(t + \widehat{d}_t \geq s)}{\mathcal{D}_{t+\widehat{d}_t}} \\
&\leq \sum_{s=1}^{T_0} \frac{\widehat{\sigma}_s^2}{\mathcal{D}_s} \\
&\leq \sum_{s=1}^{T_0} \frac{\widehat{\sigma}_s \sqrt{\mathcal{D}_s}}{K^{1/3} \log(K) \mathcal{D}_s} \\
&= \sum_{s=1}^{T_0} \frac{\widehat{\sigma}_s}{K^{1/3} \log(K) \sqrt{\mathcal{D}_s}} \\
&\leq \mathcal{O}\left(\frac{\sqrt{\mathcal{D}_{T_0}}}{K^{1/3} \log(K)}\right) = \mathcal{O}\left(\frac{\widehat{\sigma}_{\max}}{K^{2/3} \log^2(K)}\right),
\end{aligned}$$

where the second equality is by swapping the summations, the first inequality holds because $\mathcal{D}_{t+\widehat{d}_t} \geq \mathcal{D}_s$, the third inequality uses $\widehat{\sigma}_s \leq d_{\max}^s \leq \frac{\sqrt{\mathcal{D}_s}}{K^{1/3} \log K}$, and the last inequality uses Lemma 15.

The bound for $B$ is as follows

$$
\begin{aligned}
B = \sum_{t=T_0+1}^{T} \sum_{s=t+1}^{t+\widehat{d}_t} \frac{\widehat{\sigma}_s}{\mathcal{D}_{t+\widehat{d}_t} \log^2\left(\frac{\mathcal{D}_{t+\widehat{d}_t}}{\mathcal{D}_{t+\widehat{d}_t}-\mathcal{D}_t}\right)} &\leq \sum_{t=T_0+1}^{T} \sum_{s=t+1}^{t+\widehat{d}_t} \frac{\widehat{\sigma}_s}{\mathcal{D}_{t+\widehat{d}_t} \log^2\left(\frac{7K^{1/3}\log(K)\mathcal{D}_{t+\widehat{d}_t}}{\widehat{\sigma}_{\max}\sqrt{\mathcal{D}_{t+\widehat{d}_t}}}\right)} \\
&= \sum_{s=T_0+1}^{T} \sum_{t=T_0+1}^{s-1} \frac{\widehat{\sigma}_s \mathbb{1}(t+\widehat{d}_t \geq s)}{\mathcal{D}_{t+\widehat{d}_t} \log^2\left(\frac{\sqrt{7K^{1/3}\log(K)\mathcal{D}_{t+\widehat{d}_t}}}{\widehat{\sigma}_{\max}}\right)} \\
&= \sum_{s=T_0+1}^{T} \sum_{t=T_0+1}^{s-1} \frac{\widehat{\sigma}_s \mathbb{1}(t+\widehat{d}_t \geq s)}{4\mathcal{D}_{t+\widehat{d}_t} \log^2\left(\frac{49K^{2/3}\log^2(K)\mathcal{D}_{t+\widehat{d}_t}}{\widehat{\sigma}_{\max}^2}\right)} \\
&\leq \sum_{s=T_0+1}^{T} \frac{\widehat{\sigma}_s^2}{4\mathcal{D}_s \log^2\left(49K^{2/3}\log^2(K)\frac{\mathcal{D}_s}{\widehat{\sigma}_{\max}^2}\right)} \\
&\leq \widehat{\sigma}_{\max} \sum_{s=T_0+1}^{T} \frac{\widehat{\sigma}_s}{4\mathcal{D}_s \log^2\left(\frac{49K^{2/3}\log^2(K)\mathcal{D}_s}{\widehat{\sigma}_{\max}^2}\right)} \\
&\leq \widehat{\sigma}_{\max} \int_{\mathcal{D}_{T_0}}^{\mathcal{D}_T} \frac{1}{4x \log^2\left(\frac{49K^{2/3}\log^2(K)x}{\widehat{\sigma}_{\max}^2}\right)} \\
&= \widehat{\sigma}_{\max} \frac{-1}{4\log\left(\frac{49K^{2/3}\log^2(K)x}{\widehat{\sigma}_{\max}^2}\right)}\Bigg|_{\mathcal{D}_{T_0}}^{\mathcal{D}_T} = \mathcal{O}(\widehat{\sigma}_{\max}),
\end{aligned}
$$

where the first inequality follows by $\widehat{\sigma}_s \leq \widehat{\sigma}_{\max}$ and our skipping procedure that ensures $\widehat{d}_t \leq d_{\max}^t \leq \frac{\sqrt{\mathcal{D}_{t+\widehat{d}_t}}}{K^{1/3}\log K}$, the second equality is by swapping the summations, the second inequality follows by $\mathcal{D}_{t+\widehat{d}_t} \geq \mathcal{D}_s$ and $\sum_{t=1}^{s-1} \mathbb{1}(t+\widehat{d}_t \geq s) = \widehat{\sigma}_s$, the last inequality follows by Lemma 15, and the last equality uses $\int \frac{1}{x\log^2(x/\widehat{\sigma}_{\max}^2)} dx = \frac{-1}{\log(x/\widehat{\sigma}_{\max}^2)}$.

**Bound the Second Summation:** The bound for the second summation follows the same approach, but it requires additional care due to existence of $\sigma_{\max}^t$ in it. Let $T_0$ to be the time satisfying $\sqrt{\mathcal{D}_{T_0}} = \frac{\widehat{\sigma}_{\max}}{K^{1/3}\log(K)}$, then using Facts 16 and 17 we have

$$
\sum_{t=1}^{T} e^{-\frac{\mathcal{D}_{t+\sigma_{\max}^t+\widehat{d}_t}}{\mathcal{D}_{t+\sigma_{\max}^t+\widehat{d}_t}-\mathcal{D}_t}} \leq \underbrace{\sum_{t=1}^{T_0} \frac{\mathcal{D}_{t+\sigma_{\max}^t+\widehat{d}_t}-\mathcal{D}_t}{\mathcal{D}_{t+\sigma_{\max}^t+\widehat{d}_t}}}_{A} + \underbrace{\sum_{t=T_0+1}^{T} \frac{\mathcal{D}_{t+\sigma_{\max}^t+\widehat{d}_t}-\mathcal{D}_t}{\mathcal{D}_{t+\sigma_{\max}^t+\widehat{d}_t} \log^2\left(\frac{\mathcal{D}_{t+\sigma_{\max}^t+\widehat{d}_t}}{\mathcal{D}_{t+\sigma_{\max}^t+\widehat{d}_t}-\mathcal{D}_t}\right)}}_{B}.
$$

For $A$ we give the following bound

$$A = \sum_{t=1}^{T_0} e^{-\frac{\mathcal{D}_{t+\sigma_{\max}^t+\widehat{d}_t}}{\mathcal{D}_{t+\sigma_{\max}^t+\widehat{d}_t} - \mathcal{D}_t}} \leq \sum_{t=1}^{T_0} \frac{\mathcal{D}_{t+\sigma_{\max}^t+\widehat{d}_t} - \mathcal{D}_t}{\mathcal{D}_{t+\sigma_{\max}^t+\widehat{d}_t}}$$

$$= \sum_{t=1}^{T_0} \sum_{s=t+1}^{t+\sigma_{\max}^t+\widehat{d}_t} \frac{\widehat{\sigma}_s}{\mathcal{D}_{t+\sigma_{\max}^t+\widehat{d}_t}}$$

$$\leq \sum_{s=1}^{T_0} \sum_{t=0}^{s-1} \frac{\widehat{\sigma}_s \mathbb{1}(t + \sigma_{\max}^t + \widehat{d}_t \geq s)}{\mathcal{D}_s}$$

$$\leq \sum_{s=1}^{T_0} \frac{(2\sigma_{\max}^s + \widehat{\sigma}_{s-\sigma_{\max}^s})\widehat{\sigma}_s}{\mathcal{D}_s}$$

$$\leq \sum_{s=1}^{T_0} \frac{3\sqrt{\mathcal{D}_s}\widehat{\sigma}_s}{K^{1/3}\log(K)\mathcal{D}_s}$$

$$= \sum_{s=1}^{T_0} \frac{3\widehat{\sigma}_s}{K^{1/3}\log(K)\sqrt{\mathcal{D}_s}}$$

$$\leq \mathcal{O}\left(\frac{\sqrt{\mathcal{D}_{T_0}}}{K^{1/3}\log(K)}\right) = \mathcal{O}(\frac{\widehat{\sigma}_{\max}}{K^{2/3}\log^2(K)}),$$

where the first inequality is by Fact 16, the second inequality holds by swapping the summations and that $\mathcal{D}_{t+\sigma_{\max}^t+\widehat{d}_t} \geq \mathcal{D}_s$, third inequality use the following derivation

$$\mathbb{1}(t + \sigma_{\max}^t + \widehat{d}_t \geq s) \leq \mathbb{1}(t + \widehat{d}_t \geq s) + \mathbb{1}(s > t + \widehat{d}_t \geq s - \sigma_{\max}^t)$$
$$\leq \mathbb{1}(t + \widehat{d}_t \geq s) + \mathbb{1}(t \in [s - \sigma_{\max}^t, s-1]) + \mathbb{1}(t < s - \sigma_{\max}^t \wedge t + \widehat{d}_t \geq s - \sigma_{\max}^t),$$
$$(34)$$

the third equality is by swapping the summations, the third inequality uses $\widehat{\sigma}_s \leq d_{\max}^s \leq \frac{\sqrt{\mathcal{D}_s}}{K^{1/3}\log K}$, and finally the last inequality uses Lemma 15.

The bound for $B$ is as follows

$$
\begin{aligned}
B &= \sum_{t=T_0+1}^{T} \frac{\sum_{s=t+1}^{t+\sigma_{\max}^t + \widehat{d}_t} \widehat{\sigma}_s}{\mathcal{D}_{t+\sigma_{\max}^t + \widehat{d}_t} \log^2\left(\frac{\mathcal{D}_{t+\sigma_{\max}^t + \widehat{d}_t}}{\sum_{s=t+1}^{t+\sigma_{\max}^t + \widehat{d}_t} \widehat{\sigma}_s}\right)} \\
&\leq \sum_{t=T_0+1}^{T} \sum_{s=t+1}^{t+\sigma_{\max}^t + \widehat{d}_t} \frac{\widehat{\sigma}_s}{\mathcal{D}_{t+\sigma_{\max}^t + \widehat{d}_t} \log^2\left(\frac{7K^{1/3}\log(K)\mathcal{D}_{t+\sigma_{\max}^t + \widehat{d}_t}}{2\widehat{\sigma}_{\max}\sqrt{\mathcal{D}_{t+\sigma_{\max}^t + \widehat{d}_t}}}\right)} \\
&= \sum_{s=T_0+1}^{T} \sum_{t=T_0+1}^{s-1} \frac{\widehat{\sigma}_s \mathbb{1}(t+\sigma_{\max}^t + \widehat{d}_t \geq s)}{\mathcal{D}_{t+\sigma_{\max}^t + \widehat{d}_t} \log^2\left(\frac{3K^{1/3}\log(K)\sqrt{\mathcal{D}_{t+\sigma_{\max}^t + \widehat{d}_t}}}{\widehat{\sigma}_{\max}}\right)} \\
&= \sum_{s=T_0+1}^{T} \sum_{t=T_0+1}^{s-1} \frac{4\widehat{\sigma}_s \mathbb{1}(t+\sigma_{\max}^t + \widehat{d}_t \geq s)}{\mathcal{D}_{t+\sigma_{\max}^t + \widehat{d}_t} \log^2\left(\frac{9K^{2/3}\log^2(K)\mathcal{D}_{t+\sigma_{\max}^t + \widehat{d}_t}}{\widehat{\sigma}_{\max}^2}\right)} \\
&\leq \sum_{s=T_0+1}^{T} \frac{4(2\sigma_{\max}^s + \widehat{\sigma}_{s-\sigma_{\max}^s})\widehat{\sigma}_s}{\mathcal{D}_s \log^2\left(\frac{\mathcal{D}_s}{4\widehat{\sigma}_{\max}^2}\right)} \\
&\leq \widehat{\sigma}_{\max} \sum_{s=T_0+1}^{T} \frac{12\widehat{\sigma}_s}{\mathcal{D}_s \log^2\left(\frac{9K^{2/3}\log^2(K)\mathcal{D}_s}{\widehat{\sigma}_{\max}^2}\right)} \\
&\leq \widehat{\sigma}_{\max} \int_{\mathcal{D}_{T_0}}^{\mathcal{D}_T} \frac{12}{x \log^2\left(\frac{9K^{2/3}\log^2(K)x}{\widehat{\sigma}_{\max}^2}\right)} \\
&= \widehat{\sigma}_{\max} \frac{-12}{\log\left(\frac{9K^{2/3}\log^2(K)x}{\widehat{\sigma}_{\max}^2}\right)}\Bigg|_{\mathcal{D}_{T_0}}^{\mathcal{D}_T} = \mathcal{O}(\widehat{\sigma}_{\max}),
\end{aligned}
$$

where the first inequality is due to our skipping procedure that ensures $\max\left\{\sigma_{\max}^t, \widehat{d}_t\right\} \leq d_{\max}^t \leq \sqrt{\mathcal{D}_{t+\sigma_{\max}^t + \widehat{d}_t}}$, the second equality is by swapping the summations, the second inequality follows by $\mathcal{D}_{t+\widehat{d}_t} \geq \mathcal{D}_s$ and (34), the last inequality follows by Lemma 15, and the last equality uses $\int \frac{1}{x \log^2(x/\widehat{\sigma}_{\max}^2)} dx = \frac{-1}{\log(x/\widehat{\sigma}_{\max}^2)}$. ∎

# E  A proof of Lemma 5

*Proof.* We use the term *free round* to refer to a round $r$ such that $\upsilon_r^{new}$ is zero. By applying induction on the time step $t$, we show that if the algorithm is currently at time $t$ and intends to rearrange the $\upsilon_t$ arrivals, there exist $\upsilon_t$ free rounds in the interval $[t, t + \sigma_{\max}^t - \widehat{\sigma}_t + \upsilon_t]$ to which the algorithm can push the arrivals. This ensures that the arrival from round $s$, will be rearranged to round $\pi(s) \geq s + \widehat{d}_s$, such that $\pi(s) - (s + \widehat{d}_s) \leq \sigma_{\max}^t$. To this end, we assume the induction assumption holds for all $r < t$, and then proceed with induction step for $t$.

**Induction Base:**
The induction base corresponds to the first arrival time, denoted as $t_0$. At this time step, all $\upsilon_{t_0}$ arrivals can be rearranged to the free rounds in the interval $[t_0, t_0 + \upsilon_{t_0} - 1]$, which is a subset of $[t_0, t_0 + \sigma_{\max}^{t_0} - \widehat{\sigma}_{t_0} + \upsilon_{t_0} - 1]$. Therefore, the induction base holds.

**Induction step:**
Assume that we are at round $t$, and our aim is to rearrange the arrivals of round $t$. We define $t_1$ as the last occupied round, where $t_1 \geq t$. So it suffices to prove $t_1 - t \leq \sigma_{\max}^t - \widehat{\sigma}_t$. We first note that since the algorithm is greedy, all rounds $t, t+1, \ldots, t_1 - 1$ must also be occupied by some arrivals from the past.

Let $t_0 < t$ be the first round where one of its arrivals has been rearranged to $t$, and let $v'_{t_0}$ be the number of arrivals at time $t_0$ that are rearranged to some rounds before $t$. Then by induction assumption we know

$$t - t_0 \leq \sigma_{\max}^{t_0} - \widehat{\sigma}_{t_0} + v'_{t_0} + 1 = \sigma_{\max}^{t_0} - \sum_{r=1}^{t_0-1} \mathbb{1}(r + \widehat{d}_r \geq t_0) + v'_{t_0} + 1. \tag{35}$$

On the other hand, by the choice of $t_0$, each occupied round $t, t+1, \ldots, t_1$ must be occupied by exactly one arrival among the arrivals of rounds $t_0, \ldots, t-1$, except for the $v'_t$ arrivals of $t_0$ that are rearranged to some rounds before $t$. So we have

$$
\begin{aligned}
t_1 - t + 1 &\leq \sum_{r=1}^{t-1} \mathbb{1}(t_0 \leq r + \widehat{d}_r \leq t - 1) - v'_{t_0} \\
&= \sum_{r=1}^{t_0-1} \mathbb{1}(t_0 \leq r + \widehat{d}_r \leq t - 1) + \sum_{r=t_0}^{t-1} \mathbb{1}(t_0 \leq r + \widehat{d}_r \leq t - 1) - v'_{t_0} \\
&= \sum_{r=1}^{t_0-1} \mathbb{1}(t_0 \leq r + \widehat{d}_r \leq t - 1) + t - t_0 - \sum_{r=t_0}^{t-1} \mathbb{1}(r + \widehat{d}_r \geq t) - v'_{t_0},
\end{aligned}
$$

where the second equality holds because $\sum_{r=t_0}^{t-1} \mathbb{1}(r + \widehat{d}_r \geq t_0) = t - t_0$. We use (35) to bound $t - t_0$ in the above inequality and get

$$
\begin{aligned}
t_1 - t &\leq \sigma_{\max}^{t_0} + \sum_{r=1}^{t_0-1} \mathbb{1}(t_0 \leq r + \widehat{d}_r \leq t - 1) - \sum_{r=1}^{t_0-1} \mathbb{1}(r + \widehat{d}_r \geq t_0) - \sum_{r=t_0}^{t-1} \mathbb{1}(r + \widehat{d}_r \geq t) \\
&= \sigma_{\max}^{t_0} - \sum_{r=1}^{t_0-1} \mathbb{1}(r + \widehat{d}_r \geq t) - \sum_{r=t_0}^{t-1} \mathbb{1}(r + \widehat{d}_r \geq t) \\
&= \sigma_{\max}^{t_0} - \sum_{r=1}^{t-1} \mathbb{1}(r + \widehat{d}_r \geq t) \leq \sigma_{\max}^t - \widehat{\sigma}_t, \tag{36}
\end{aligned}
$$

where the last inequality follows by the fact that $\{\sigma_{\max}^r\}_{r \in [T]}$ is a non-decreasing sequence. So if the algorithm rearranges the $v_t$ arrivals at round $t$ to rounds $t_1 + 1, \ldots, t_1 + v_t$, then, using the inequality (36), we can conclude that these rounds fall within the interval $[t, t + \sigma_{\max}^t - \widehat{\sigma}_t + v_t]$. ∎

## F  Adversarial bounds with $d_{\max}$ cannot benefit from skipping

In this section we show that adversarial regret bounds that involve terms that are linear in $d_{\max}$, such as the bounds of Masoudian et al. (2022), cannot benefit from skipping. We prove the following lemma.

**Lemma 18.**
$$\sqrt{D} \leq \min_{\mathcal{S}} \left(|\mathcal{S}| + \sqrt{D_{\bar{\mathcal{S}}}}\right) + d_{\max}.$$

*Proof.* For any split of the rounds $[T]$ into $\mathcal{S}$ and $\bar{\mathcal{S}}$ we have

$$D = D_{\bar{\mathcal{S}}} + D_{\mathcal{S}} \leq D_{\bar{\mathcal{S}}} + |\mathcal{S}| d_{\max} \leq D_{\bar{\mathcal{S}}} + |\mathcal{S}|^2 + d_{\max}^2.$$

Thus

$$\sqrt{D} \leq \sqrt{D_{\bar{\mathcal{S}}} + |\mathcal{S}|^2 + d_{\max}^2} \leq |\mathcal{S}| + \sqrt{D_{\bar{\mathcal{S}}}} + d_{\max},$$

and since the above holds for any $\mathcal{S}$, we obtain the statement of the lemma. ∎

We remind that skipping allows to replace a term of order $\sqrt{D}$ by a term of order $\min_{\mathcal{S}} \left(|\mathcal{S}| + \sqrt{D_{\bar{\mathcal{S}}}}\right)$ (for simplicity we ignore factors dependent on $K$). Thus, it may potentially replace a bound of order $\sqrt{D} + d_{\max}$ by a bound of order $\min_{\mathcal{S}} \left(|\mathcal{S}| + \sqrt{D_{\bar{\mathcal{S}}}}\right) + d_{\max}$, but since by the lemma $\min_{\mathcal{S}} \left(|\mathcal{S}| + \sqrt{D_{\bar{\mathcal{S}}}}\right) + d_{\max} = \Omega(\sqrt{D})$, this would not improve the order of the bound.

# G  Details of the Adversarial Analysis

The only difference between our algorithm and the algorithm of Zimmert and Seldin (2020) is the implicit exploration and the slightly modified skipping rule. Let $\ell_t$ be the original loss sequence, then the adversary can create an adaptive sequence $\widetilde{\ell}_t$ that forces the player to play according to the implicit exploration rule by simply down-scaling all the losses by

$$\widetilde{\ell}_{ti} = \frac{x_{ti}\ell_{ti}}{\max\left\{x_{t,i}, \lambda_{t,t+\widehat{d}_t}\right\}} \ .$$

Our regret bound decomposes now into

$$\overline{Reg}_T = \max_{i_T^*} \mathbb{E}\left[\sum_{t=1}^T \langle x_t, \ell_t \rangle - \ell_{t,i_T^*}\right]$$

$$\leq \max_{i_T^*} \mathbb{E}\left[\sum_{t=1}^T \left\langle x_t, \widetilde{\ell}_t \right\rangle - \widetilde{\ell}_{t,i_T^*}\right] + \mathbb{E}\left[\sum_{t=1}^T \left\langle x_t, \ell_t - \widetilde{\ell}_t \right\rangle\right] \ .$$

For the second term we have

$$\sum_{t=1}^T \left\langle x_t, \ell_t - \widetilde{\ell}_t \right\rangle \leq \sum_{i=1}^K \sum_{t=1}^T (1 - \frac{x_{ti}}{x_{ti} + \lambda_{t,t+\widehat{d}_t}})x_{ti} \leq K \sum_{t=1}^T \lambda_{t,t+\widehat{d}_t} \ ,$$

which can be controlled via Lemma 4.

The first term is bounded by Zimmert and Seldin (2020, Theorem 3) (since the player plays their algorithm on the modified loss sequence) by

$$\max_{i_T^*} \mathbb{E}\left[\sum_{t=1}^T \langle x_t, \ell_t \rangle - \ell_{t,i_T^*}\right] \leq 4\sqrt{KT} + \sum_{t=1}^T \gamma_t \widehat{\sigma}_t + \gamma_T^{-1} \log K + S^*$$

$$\leq 4\sqrt{KT} + \sum_{t=1}^T \frac{\widehat{\sigma}_t\sqrt{\log K}}{7\sqrt{\mathcal{D}_t}} + 7\sqrt{\mathcal{D}_T \log K} + S^*$$

$$= 4\sqrt{KT} + \sqrt{\log K}\sum_{t=1}^T \frac{\mathcal{D}_t - \mathcal{D}_{t-1}}{7\sqrt{\mathcal{D}_t}} + 7\sqrt{\mathcal{D}_T \log K} + S^*$$

$$\leq 4\sqrt{KT} + \frac{2\sqrt{\log K}}{7}\sum_{t=1}^T \sqrt{\mathcal{D}_t} - \sqrt{\mathcal{D}_{t-1}} + 7\sqrt{\mathcal{D}_T \log K} + S^*$$

$$= 4\sqrt{KT} + \frac{51}{7}\sqrt{\mathcal{D}_T \log K} + S^*$$

$$\leq 4\sqrt{KT} + \frac{51}{7}\min_{\mathcal{S}\subseteq[T]}\left\{|\mathcal{S}| + \sqrt{\mathcal{D}_{\bar{\mathcal{S}}} \log K}\right\} + S^*,$$

where the first equality uses the definition of $\gamma_t$, the third inequality follows by $\forall a, b > 0: \frac{a-b}{\sqrt{a}} \leq 2(\sqrt{a} - \sqrt{b})$, and the last inequality uses the following lemma

**Lemma 19.** *The skipping technique guarantees the following bound*

$$\sqrt{\mathcal{D}_T K^{\frac{2}{3}} \log K} \leq \min_{\mathcal{S}\subseteq[T]}\left\{|\mathcal{S}| + \sqrt{\mathcal{D}_{\bar{\mathcal{S}}} K^{\frac{2}{3}} \log K}\right\} \ .$$

Combining the bounds on the first and the second terms provides the regret bound in Section 5.2. It only remains to provide a proof for Lemma 19.

*Proof of Lemma 19.* For any $t \in [T]$ we have $\widehat{d}_t \leq \sqrt{\mathcal{D}_T/(49K^{\frac{2}{3}}\log(K))}$, therefore for any $R \subset [T]$:

$$\sum_{t\in[T]\setminus R} d_t \geq \sum_{t\in[T]\setminus R} \widehat{d}_t \geq \mathcal{D}_T - |R|\sqrt{\mathcal{D}_T/(49K^{\frac{2}{3}}\log(K))}$$

Hence we can dereive the following lower bound,

$$\min_{R\subseteq[T]} |R| + \sqrt{\sum_{s\in[T]\setminus R} d_s K^{\frac{2}{3}} \log(K)} \geq \min_{r\in\left[0,\sqrt{49\mathcal{D}_T K^{\frac{2}{3}} \log(K)}\right]} r + \sqrt{\mathcal{D}_T K^{\frac{2}{3}} \log(K) - \frac{1}{7}r\sqrt{\mathcal{D}_T K^{\frac{2}{3}} \log(K)}}$$

$$\geq \sqrt{\mathcal{D}_T K^{\frac{2}{3}} \log(K)},$$

where the second inequality uses the concavity in $r$.

# H   A Bound on $S^*$

Next, we reason about the nature of skips. The following lemma is an adaptation of Zimmert and Seldin (2020, Lemma 5) to our skipping threshold. To this end we provide two lemmas and then conclude then proof.

**Lemma 20.** *Algorithm 1 will not skip more than 1 point at a time.*

*Proof.* We prove the lemma by contradiction. Assume that $s_1, s_2$ are both deactivated at time $t$. W.l.o.g. let $s_2 \leq s_1 - 1$. Skipping of $s_1$ at time $t$ means $t - s_1 \geq \sqrt{\mathcal{D}_t/(K^{\frac{2}{3}} \log(K))} \geq \sqrt{\mathcal{D}_{t-1}/(K^{\frac{2}{3}} \log(K))}$. At the same time we assumed $t - 1 - s_2 \geq t - s_1$, which means that $s_2$ would have been deactivated at round $t - 1$ or earlier. ∎

Recall that $\widehat{d}_t$ is the contribution of a timestep $t$ to the sum $\mathcal{D}_T$. Let $(t_1, \ldots, t_{S^*})$ be an indexing of $\mathcal{S}$ and $c = 49K^{\frac{2}{3}} \log(K)$. We bound the number of skips by

$$S^* \leq 2c\widehat{d}_{t_S^*}. \tag{37}$$

The above bound together with the fact that incurred delay $\widehat{d}_{t_S^*}$ must be less than the the skipping threshold and the maximal delay $d_{\max}$ give us

$$S^* \leq \mathcal{O}\left(K^{\frac{2}{3}} \log K \widehat{d}_{t_S^*}\right)$$

$$\leq \mathcal{O}\left(\min\left\{d_{\max} K^{\frac{2}{3}} \log K, \sqrt{\mathcal{D}_T K^{\frac{2}{3}} \log K}\right\}\right)$$

$$\leq \mathcal{O}\left(\min\left\{d_{\max} K^{\frac{2}{3}} \log K, \min_{\mathcal{S}\subseteq[T]}\left\{|\mathcal{S}| + \sqrt{\mathcal{D}_{\bar{\mathcal{S}}} K^{\frac{2}{3}} \log K}\right\}\right\}\right),$$

where the last inequality follows by Lemma 19. ∎

*Proof of bound* (37). By Lemma 20 we skip at most one outstanding observation per round. Thus, we have that

$$\widehat{d}_{t_m} \geq \sqrt{\mathcal{D}_{t_m + \widehat{d}_{t_m}}/c} \geq \sqrt{\sum_{i=1}^{m} \widehat{d}_{t_i}/c} = \frac{\sqrt{\widehat{d}_{t_m} + \sum_{i=1}^{m-1} \widehat{d}_{t_i}}}{\sqrt{c}}.$$

By solving the quadratic inequality in $\widehat{d}_{t_m}$ we obtain

$$\widehat{d}_{t_m} \geq \frac{1 + \sqrt{1 + 4c\sum_{i=1}^{m-1} \widehat{d}_{t_i}}}{2c}.$$

Now we prove by induction that $\widehat{d}_{t_m} \geq \frac{m}{2c}$. The induction base holds since $\widehat{d}_{t_1} = 1$. For the inductive step we have

$$\widehat{d}_{t_m} \geq \frac{1 + \sqrt{1 + 4c\sum_{i=1}^{m-1} \widehat{d}_{t_i}}}{2c} \geq \frac{1 + \sqrt{1 + m(m-1)}}{2c} \geq \frac{m}{2c}.$$

Then the induction step is satisfied. ∎

