# OpenReview forum: "A Best-of-both-worlds Algorithm for Bandits with Delayed Feedback with Robustness to Excessive Delays"
_NeurIPS.cc/2024/Conference — NeurIPS 2024 poster_

### Official Review · Reviewer_bmXx · 2024-06-13

**Soundness:** 3
**Presentation:** 2
**Contribution:** 3
**Rating:** 6
**Confidence:** 3

**Summary:**

This paper studies the best-of-both-worlds (BOBW) algorithms for MAB with delayed feedback. Compared to previous results, it eliminates the need for prior knowledge on maximum delay $d_{\text{max}}$, and the regret scales with the number of “outstanding observations” ($\sigma_{\text{max}}$) rather than with the delay length as in the previous works. Technique-wise, this is achieved through a new round skipping rule, a new implicit exploration loss estimator, and advanced analysis.

**Strengths:**

The regret guarantee is significant in the sense that it scales with the number of missing information, rather than how much it is delayed as in previous work. In the latter case, even if there’s only one single delay, it incurs linear regret if this delay is $T$, which however is safeguarded by the regret guarantee in this paper. Such phenomenon has been achieved soly in either stochastic or adversarial regime before, but not in BOBW.  To achieve this, the authors propose new techniques in both algorithm design and regret analysis.

**Weaknesses:**

1. I feel that the words "distribution drift/shift" comes from nowhere in the introduction. Its meaning is not clear to me in the context of delayed bandits, but no sufficient explanation is provided. A reader may not figure it out until the Analysis Section.
2. The authors use both “multiarm” and “multi-arm” inconsistently. The latter is commonly used.
3. The contribution summary may need polishing. Points 2-4 (Line 118) seems too  detailed. Currently they are not informative, they could be merged, or not even necessarily included in the contribution summary in my opinion.
4. The author may consider elaborating more on Line 181-183, i.e., why the additional $K^{1/3}$ factor is a price for BOBW guarantee (in the analysis). This could be an interesting point to BOBW researchers.
5. When introducing Lemma 2 for the first time (around Line 202), it would be good to briefly talk about what are the key elements to achieve it, rather than let the readers figure it out in the long proof later. Is it due to the analysis (virtual rearrangement) only, or also the skipping rule and/or implicit exploration?
6. These two works on feedback-delayed bandits should be relevant, but they are not discussed in this paper. One in stochastic regime [1] and the other in adversarial regime [2].

References:

[1] Yang, Yunchang, Han Zhong, Tianhao Wu, Bin Liu, Liwei Wang, and Simon Shaolei Du. "A Reduction-based Framework for Sequential Decision Making with Delayed Feedback." In Thirty-seventh Conference on Neural Information Processing Systems. 2023.

[2] van der Hoeven, Dirk, Lukas Zierahn, Tal Lancewicki, Aviv Rosenberg, and Nicoló Cesa-Bianchi. "A Unified Analysis of Nonstochastic Delayed Feedback for Combinatorial Semi-Bandits, Linear Bandits, and MDPs." In The Thirty Sixth Annual Conference on Learning Theory, pp. 1285-1321. PMLR, 2023.

**Questions:**

1. Why do some citations have no year? E.g., “analysis of Zimmert and Seldin” in Line 58

2. Is the word “outstanding observations” existing in the delayed-feedback literature or named for the first time by the authors? It is just the number of missing observations, right? If so, I personally just feel that the wording is not very informative/natural.

3. Is the idea of “skipping” reflected only through the regret analysis, or both the algorithm design and analysis? From Algorithm 1, I think it is the latter, but Line 86-89 seems to mean the former.

4. Since the BOBW guarantee is obtained through the self-bounding trick, can we directly obtain a regret bound for the intermediate $C$-corrupted regime?

**Limitations:**

Yes

---

> ### Author Rebuttal · Authors · 2024-08-06
>
> **Weaknesses:**
>
>
> 1. We will add a clarification around Lines 48-49, where the term is first used. Technically, the distribution drift is the ratio $x_{t+d_t,i} / x_{t,i}$ of the probability of playing action $i$ at round $t+d_t$, when the observation arrives, to the probability it had when it was played at round $t$.
> 2. Thanks, we will fix it.
>
> 3. Thanks for the suggestion.
>
> 4. We had to decrease the skipping threshold of Zimmert and Seldin (2020) to control the distribution drift that is due to the loss shift (see Appendix B.2 in the supplementary material). More explicitly, $K^{1/3}$ comes from a bound on the weighted average of loss estimates in Equation (25) that is bounded in Lemma 9. For now we do not know how to remove this factor, but we expect that it might require a different analysis of the distribution drift.
>
> 5. The core of Lemma 2, which relates drifted regret to the actual regret, is based on Lemma 3, which controls the distribution drift using implicit exploration and skipping. The factor of 1/4 in front of $\overline{Reg}\_T$ in Lemma 2 and the summation of implicit exploration terms come from Lemma 3. Virtual rearrangement of arrivals is needed in the proof of Lemma 2, because at the moment we have no other way to analyze multiple simultaneous arrivals. Lemma 5, which analyzes Algorithm 2 for virtual rearrangements, shows that the additional delay caused by the rearrangements is small (limited by $\sigma_{\max}^t$).
>
> 6. Thanks for the references.
> ***
> **Questions:**
> 1. It is a common practice to use \citeauthor{...} command on repeated mentions of the same paper within a short text span distance, whenever it does not lead to confusion which work is being mentioned. We hope it was not confusing in Line 58. We can add a year if necessary.
>
> 2. The term “outstanding observations” was introduced by Zimmert and Seldin (2020). An observation is considered “outstanding” from the moment an action from which it originates is played to the moment it is revealed to the algorithm.
>
> 3. Skipping is done both in the algorithm and in the analysis.
>
> 4. Yes, it is straightforward to obtain a bound for $C$-corrupted regime.

---

> > ### Comment · Reviewer_bmXx · 2024-08-07
> >
> > I'd like to thank the authors for the response. For now, I do not have any other concerns, but I suggest that the authors could improve the writing in furture versions, including (but not limited to):
> >
> > 1. Clarifying "distribution drift/shift", as the meaning here is different from that in supervised learning (which is more well-known).
> > 2. Improving the contribution summary.
> > 3. Giving more details about the extra $K^{1/3}$ factor.
> > 4. Overall, making a clearer roadmap about the analysis. Currently, it's a bit too dense in my opinion.
> > 5. Including those two papers I mentioned, and some discussions (especially [2], the one for adversarial environment and hence is related to FTRL/OMD, why the analysis therein fails to obtain the guarantees here).
> >
> > I keep my score unchanged as of now and lean towards acceptance.

---

> > > ### Author Response · Authors · 2024-08-08
> > > **Response**
> > >
> > > We thank the reviewer for additional input, which we will incorporate in the final version.
> > >
> > > A clarification concerning [2]: the work [2] focuses on adversarial setting only, and they assume that either the delays $d$ are fixed, or the total delay $D$ *and* the maximal delay $d_{\max}$ are known. The focus of our work is on BOBW bounds, on handling excessive delays with no prior knowledge of delays, and on eliminating the dependence on $d_{\max}$. So the challenges addressed by us are very different from theirs. We also note that [2] explicitly state in their Discussion that skipping cannot be applied with their approach, meaning that it is unable to handle delay outliers, and that their bounds depend on $d_{\max}^2$, meaning that even a single delay of order $\sqrt{T}$ would make their bounds linear in $T$.

---

### Official Review · Reviewer_9WBQ · 2024-06-30

**Soundness:** 3
**Presentation:** 3
**Contribution:** 3
**Rating:** 6
**Confidence:** 3

**Summary:**

This paper proposes a new best-of-both-world algorithm for bandits with a delayed feedback model. The results simultaneously achieve the latest upper bounds for delayed feedback in stochastic and adversarial (up to a factor). The algorithm design utilizes an implicit exploration estimator and the skipping set. The analysis contains new technical methods for handling the relationship between drifted and standard regrets.

**Strengths:**

The reviewer thinks this paper is a nice piece of theoretical work. It has several strengths as follows,
1. This paper's introduction concisely and clearly discusses this work and its relation (both result and theoretical tool aspects) to previous work. This is very helpful for readers to understand the research background and new contribution.
2. The best-of-both-world theoretical results of this paper combine the known best stochastic and adversarial upper bounds (with a $K^{1/3}$) factor for bandits with delay and improve upon the latest results on the same topic.

**Weaknesses:**

The reviewer is uncertain whether this paper reaches the NeurIPS bar for the following weaknesses reasons:
1. The algorithm design itself needs to be more novel. The only new component the reviewer notices is the implicit exploration estimator, which was also related to prior works in adversarial bandits.
2. The algorithm design is not optimized. For example, the skipping set $\mathcal{S_{t}}$ is always expanding, which is counterintuitive: (1) As the threshold $d^t_{\text{max}}$ is non-decreasing, there should exist observations, previous belongs to the skipping set, with delay less than the increased new threshold. Why does the algorithm design not add these skipped observations to the non-skipped set? (2) Even for observations belonging to the skipping set, when their observations are finally revealed, why does the algorithm not consider the kind of observations? If considered by the decision-making, both types of observations can help improve the algorithm's performance. Even if this modification cannot reduce the regret upper bound, it should be able to improve the algorithm's empirical performance.

**Questions:**

1. The reviewer needs clarification on some proof steps. For example, by the definition of the drift regret, the benchmark is the estimated loss $\hat \ell_{i^*}$, but when it comes to the stability-penalty decomposition at the beginning of Appendix A, the benchmark becomes the actual loss $\ell_{i^*}$, which looks inconsistent.

**Limitations:**

Yes

---

> ### Author Rebuttal · Authors · 2024-08-06
>
> > The algorithm design itself needs to be more novel
>
> It is very common for BOBW algorithms to bear close resemblance to the adversarial algorithms they are derived from, because it allows inheritance of the adversarial regret guarantees. For example, the EXP3++ algorithm (Seldin and Slivkins, 2014) only makes a small modification to the exploration distribution of the well-known EXP3 algorithm, and the Tsallis-INF algorithm (Zimmert and Seldin, 2021) is essentially identical to the INF algorithm designed by Audibert and Bubeck (2010) for adversarial bandits. The major contribution of many BOBW papers, including ours, is the stochastic analysis that stays within the predefined framework of the parent adversarial algorithm, which only allows minor algorithmic modifications due to the necessity to preserve the adversarial regret guarantees. The preservation of the algorithm design is, therefore, a desired feature. The value and technical sophistication of these contributions, including ours, should not be underestimated though. We introduced several important algorithmic and analytical tools:
> 1. We are the first to control the distribution drift without altering the learning rates, by using implicit exploration and skipping (Lemma 3). Prior work controlled the drift by damping the learning rates using knowledge of $d_{\max}$, but this technique adds $d_{\max}$ linearly to the regret bound and fails in presence of just a single outlier of order $d_{\max}$.
> 2. Implicit exploration design is very delicate, because it has to be sufficiently large to control the drift, but at the same time sufficiently small, because it adds up linearly to the bound. We have designed an implicit exploration scheme that does not ruin neither the stochastic nor the adversarial bound (Lemma 4). Note that our implicit exploration scheme depends on the delays, but not the losses. Therefore, it has a great potential to be useful in other settings with delayed feedback.
> 3. We have also derived the first approach to relate drifted and non-drifted regret in presence of delay outliers (Lemma 2). This contribution involves introduction of the greedy rearrangement algorithm (Algorithm 2) and drift control by Lemma 3. Prior work has only allowed to relate drifted regret and the actual regret, when the delays were bounded by $d_{\max}$, and it was adding $d_{\max}$ linearly to the regret bound. Thus, it was failing in presence of even a single delay of order $T$.
>
> We believe that our contribution has the potential to bring a transformative effect to the field of BOBW learning with delayed feedback.
>
> ***
> > The algorithm design is not optimized
>
> - **Concerning (1):** when an observation is processed at time $t$ it modifies the playing distribution $x_t$ at time $t$. If at time $t$ we would decide to go back to some old observations, the actual delay from the moment the action was played to the moment it is reconsidered would remain above the updated threshold, so it would not be possible to add it back.
>
> - **Concerning (2):** while it feels wasteful to drop observations, we note that the cardinality of the skipped set is smaller than the regret bound, which is in turn sublinear in the total number of observations. Processing of skipped observations would add a lot of pain in the analysis, but add nothing in terms of the bound (because of matching lower bounds), and almost nothing in terms of empirical performance, because of the small size of the skipped set relative to the total number of observations.
>
> ***
> **Q1:** We are sorry, it is a typo. It should be $\hat \ell_{t, i_{T}^{*}}$ at the end of the display between Lines 328 and 329 in Appendix A.
> ***
> **References**
> Y. Seldin and A. Slivkins. One practical algorithm for both stochastic and adversarial bandits. ICML, 2014.
> J-Y. Audibert and S. Bubeck. Regret bounds and minimax policies under partial monitoring. JMLR, 2010.

---

> > ### Comment · Reviewer_9WBQ · 2024-08-08
> >
> > The reviewer thanks the authors for their responses. The reviewer will maintain their score for now.

---

### Official Review · Reviewer_oEfL · 2024-07-04

**Soundness:** 3
**Presentation:** 2
**Contribution:** 3
**Rating:** 6
**Confidence:** 3

**Summary:**

This paper studied the best of both world algorithms for arbitrary delayed feedback. The proposed algorithm does not require prior knowledge of maximum delay $d_{\max}$ and avoids its linear dependence in the regret bound. To this end, they proposed the implicit exploration that works for the best-of-both-worlds guarantees. The key idea is to relate the number of outstanding observations $\sigma_{\max}$ so as not to rely on the boundness of delays.

**Strengths:**

- The first proposal of implicit exploration for BoBW settings. The algorithm is based on FTRL with a hybrid regularizer where a skipping technique is employed, which becomes crucial for the BoBW guarantee.

- When the maximal number of outstanding observations $\sigma_{\max}$ is smaller than maximum delay $d_{\max}$, the proposed method provided a much better bound than [Masoudian et al. 2022].

**Weaknesses:**

I only see it for writing.
The current main paper more focus on the explanation of the result comparison with [Masoudian et al. 2022]. Although the authors detailed a summary of technical proof in the main text, a more intuitive, and higher-level explanation of algorithmic parts would be appreciated when introducing the proposed technical scheme. For example, the paper does not provide a detailed explanation of the selection of regularizers or learning rates intuitively. For example, what makes analysis challenging due to aiming BoBW bounds could be more highlighted. I guess the additional challenge for dealing with stochastic cases, from [Zimmer and Seldin, 2020] that already uses the FTRL algorithm with a hybrid regularizer, has already been addressed by [Masoudian et al. 2022], but by missing these explanations, it would be hard for readers to see the clear picture of inherently technical challenges of BoBW algorithm for delayed feedback.

**Questions:**

How can we interpret implicit exploration term $\lambda_{s,t}$? (Is this inspired by controlling the bias term due to an implicit exploration scheme in order to make it applicable to a best-of-both-worlds setting?)

**Limitations:**

n.a

---

> ### Author Rebuttal · Authors · 2024-08-06
>
> **A detailed explanation of the selection of regularizers or learning rates:** The learning rates and regularizers were taken directly from Zimmert and Seldin (2020), because we had to have the adversarial regret guarantee. The intuition behind the choice of regularizers is that the adversarial regret bound, $\sqrt{KT} + \sqrt{dT\log K}$ for fixed delays $d$, and the construction of the matching lower bound combine a lower bound for bandits to show the necessity of the $\sqrt{KT}$ term, and a full information lower bound to show the necessity of the $\sqrt{dT / logK}$ term. The negative Tsallis-entropy regularizer with power 1/2 is optimal in the bandit setting, and the negative entropy regularizer is optimal in the full information setting, and the combination achieves the optimal regret bound in the delayed feedback setting. The learning rate $\eta_t$ is the standard learning rate for bandits, and the learning rate $\gamma_t$ is the standard full information learning rate. We refer to Zimmert and Seldin (2020) for further details. The primary challenge addressed by our work is that these regularizers and learning rates also work in the stochastic setting. We make a small adjustment of $\gamma_t$ by a constant due to adjusted skipping.
>
> ***
> **What makes analysis challenging due to aiming BoBW bounds:** First, we want to emphasize that our focus is on BoBW analysis in presence of delay outliers, and that the work of Masoudian et al. (2022) cannot handle delay outliers, because even a single delay outlier of order $T$ renders both their stochastic and adversarial regret bound linear. Concerning analytical challenges: the stochastic part of our analysis is based on control of the distribution drift (the ratio $x_{t+d_t,i} / x_{t,i}$ of the probability of playing action $i$ at round $t+d_t$, when the observation arrives, to the probability it had when it was played at round $t$). Control of the distribution drift is the most commonly used approach in bandits with delayed feedback, also used by Masoudian et al. (2022). So far the only way to control the distribution drift was by damping the learning rate, but it only applies when the maximal delay $d_{\max}$, which corresponds to the control range, is known, and it adds $d_{\max}$ linearly to the regret bound. Therefore, this technique fails in presence of delay outliers. We are the first to achieve control of the distribution drift in presence of delay outliers. In order to achieve it we introduced several important algorithmic and analytical tools:
> 1. We are the first to control the distribution drift without altering the learning rate. Our control is based on a combination of implicit exploration, which is used to control the drift due to the change of regularizer, and skipping, which is used to control the drift due to the loss shift (Lemma 3).
> 2. The challenge in designing a successful implicit exploration scheme is that implicit exploration has to be sufficiently large to control the ratio, but at the same time sufficiently small, because it adds up linearly to the bound. We have shown that the contribution of our implicit exploration to the bound, $\sum_{t=1}^T \lambda_{t,t+\hat d_t}$, does not ruin neither the stochastic nor the adversarial bound (Lemma 4).
> 3. We have also derived the first approach to relate drifted and non-drifted regret in presence of delay outliers (Lemma 2). Prior work has only allowed to relate the two when the delays were bounded by $d_{\max}$, and it was adding $d_{\max}$ linearly to the regret bound. Thus, it was failing in presence of even a single delay of order $T$.
>
> We believe that all these tools will find additional applications in other learning settings.
> Finally, we note that the analysis of Zimmert and Seldin (2020) only applies in the adversarial setting. It is unknown whether their analysis technique, which is not based on distribution drift, can be applied in the stochastic setting.
> ***
> **Interpretation of implicit exploration terms $\lambda_{s,t}$:** The role of $\lambda_{s,t}$ is to control the distribution drift (the ratio $x_{t+d_t,i} / \max(x_{t,i}, \lambda_{t,t+d_t})$). To control the ratio it cannot be too small, but since $\sum_{t=1}^T \lambda_{t,t+\hat d_t}$ adds up linearly to the bound, it cannot be too large either. Note that $\lambda_{s,t}$ only depends on the delays, but not the losses.

---

> > ### Comment · Reviewer_oEfL · 2024-08-10
> >
> > Thank you for providing a detailed response.
> > I have read the reviews and responses from the other reviewers as well, and then I will keep my score (leaning towards acceptance).

---

### Official Review · Reviewer_2mU9 · 2024-07-13

**Soundness:** 3
**Presentation:** 2
**Contribution:** 3
**Rating:** 6
**Confidence:** 4

**Summary:**

The authors consider the multi-armed bandit problem with delayed feedback, where the loss of a chosen arm is observed several rounds later. In this setting, nearly optimal algorithms have been developed for both stochastic and adversarial environments, as well as best-of-both-worlds (BOBW) algorithms that perform well in both regimes. However, the regret upper bounds of existing BOBW algorithm by Masoudian et al. (2022) has gaps compared to the regret upper bounds of algorithms designed specifically for adversarial environments (Zimmert and Seldin, 2020) and stochastic environments (Joulani et al., 2013) and in particular the algorithm by Masoudian et al. require the prior knowledge of the maximal delay before the game starts.
To address this issue, the authors propose an algorithm that achieves a regret upper bound without knowing the maximal delay in advance.
To achieve this, the authors introduce a new framework of implicit exploration, demonstrating that this framework can be effectively combined with techniques that skip observations associated with excessive delays.

**Strengths:**

The differences from existing research are thoroughly discussed. Technically, the introduction of the implicit exploration framework to achieve BOBW regret bounds without using the maximal delay is novel. Fine-tuned adjustments are made to achieve BOBW while employing implicit exploration. Additionally, how the parameter $\lambda$ is utilized in the proof is clearly explained and discussed throughout the proof.

**Weaknesses:**

One concern is that the differences between this paper and existing research might not be sufficient. While the authors propose an algorithm that works in both adversarial and stochastic environments, the foundation for handling excessive delays seems to have been largely established by Zimmert and Seldin (2020) (and Masoudian (2022)).

I was unable to check all the proofs in detail, but they appear to be generally correct. However, there seems to be large room for improvement in the overall explanation. Many explanations are not contextualized, making the proofs difficult to read.
For instance, Lemmas 11 to 13 are directly taken from Masoudian et al. (2022). However, the algorithm in Masoudian et al. (2022) differs from the one proposed by the authors, so it is unclear whether these results are directly applicable.

**Questions:**

The authors are expected to discuss to what extent the results from Zimmert and Seldin (2020) and Masoudian et al. (2022) can be used in their study.

---

> ### Author Rebuttal · Authors · 2024-08-06
>
> **Concerning the novelty of our approach to handling excessive delays:**  First, we want to emphasize that the work of Masoudian et al. (2022) is unable to cope with excessive delays, because even a single delay of order $T$ renders their regret bound linear in both stochastic and adversarial environments. The work of Zimmert and Seldin (2020) only copes with excessive delays in the adversarial regime. It is the only work on delayed feedback and adversarial losses known to us that is not using control of the distribution drift in the analysis, but it is unknown whether their analysis technique can be applied to stochastic losses to obtain BOBW results. The stochastic part of our analysis is based on the more broadly used approach based on control of the distribution drift (the ratio $x_{t+d_t, i} / x_{t,i}$ of the probability of playing action $i$ at round $t+d_t$, when the observation arrives, to the probability it had when it was played at round $t$). So far the only way to control the distribution drift was by damping the learning rate, but it only works when the maximal delay $d_{\max}$, which corresponds to the control range, is known, and it adds $d_{\max}$ linearly to the regret bound. Therefore, this technique fails in presence of delay outliers. We are the first to achieve control of the distribution drift in presence of delay outliers. In order to achieve it we introduced several important algorithmic and analytical tools:
> 1. We are the first to control the distribution drift without altering the learning rate. Our control is based on a combination of implicit exploration, which is used to control the drift due to the change of regularizer, and skipping, which is used to control the drift due to the loss shift (Lemma 3).
> 2. The challenge in designing a successful implicit exploration scheme is that implicit exploration has to be sufficiently large to control the ratio, but at the same time sufficiently small, because it adds up linearly to the bound. We have shown that the contribution of our implicit exploration to the bound, $\sum_{t=1}^T \lambda_{t,t+\hat d_t}$, does not ruin neither the stochastic nor the adversarial bound (Lemma 4).
> 3. We have also derived the first approach to relate drifted and non-drifted regret in presence of delay outliers (Lemma 2). Prior work has only allowed to relate the two when the delays were bounded by $d_{\max}$, and it was adding $d_{\max}$ linearly to the regret bound. Thus, it was failing in presence of even a single delay of order $T$.
>
> We believe that all these tools will find additional applications in other learning settings.
>
> ***
> **Clarification concerning the use of Lemmas 11 to 13, which were taken from Masoudian et al. (2022):** We recall that $\overline{Reg}\_T= \sum_{t=1}^T x_{t,i}\Delta_i$. Substitute this definition into the inequality in Line 435. Lemmas 11 to 13 are used to bound three parts of the inequality in Line 435 by finding the worst-case choice of $x_{t,i}$ for each of the parts. Since the worst-case choice of $x_{t,i}$ is algorithm-independent, the lemmas can be applied. We will add the clarification to the paper.

---

> > ### Comment · Reviewer_2mU9 · 2024-08-10
> > **Response**
> >
> > I appreciate the authors' response. Since the authors addressed my concerns, I will raise the score from 5 to 6. However, as with bmXx, I expect improvements in the overall writing, particularly in the appendix.

---

> > > ### Author Response · Authors · 2024-08-12
> > >
> > > Thanks for raising the score. We will use input from all the reviewers to improve the writing.

---

### Decision · Program_Chairs · 2024-09-25

**Decision:**

Accept (poster)

**Comment:**

This paper considers a version of multi-armed bandit problem with delays in both stochastic and adversarial settings. The largest contribution of this paper is its scaling against very large delays. All reviewers agree on the novelty, and I recommend acceptance. The authors are recommended to revise the paper based on reviewers' comments.